# TO RELIEVE YOUR HEADACHE OF TRAINING AN MRF, TAKE ADVIL

**Chongxuan Li**[∗], **Chao Du**[∗], **Kun Xu**[∗], **Max Welling**[†], **Jun Zhu**[∗], **Bo Zhang**[∗]
{chongxuanli1991,duchao0726,kunxu.thu}@gmail.com,
M.Welling@uva.nl, {dcszj,dcszb}@mail.tsinghua.edu.cn

## ABSTRACT

We propose a black-box algorithm called *Adversarial Variational Inference and Learning* (AdVIL) to perform inference and learning in a general Markov random field (MRF). AdVIL employs two variational distributions to approximately infer the latent variables and estimate the partition function of an MRF, respectively. The two variational distributions provide an estimate of the negative log-likelihood of the MRF as a minimax optimization problem, which is solved by stochastic gradient descent. AdVIL is proven convergent under certain conditions. On one hand, compared to the contrastive divergence, AdVIL requires minimal assumptions about the model structure and can deal with a broader family of MRFs. On the other hand, compared to existing black-box methods, AdVIL provides a tighter estimate of the log partition function and achieves much better empirical results.

## 1 INTRODUCTION

Markov random fields (MRFs) find applications in a variety of machine learning areas (Krähenbühl & Koltun, 2011; Salakhutdinov & Larochelle, 2010; Lafferty et al., 2001). In particular, one famous example is conditional random fields (Lafferty et al., 2001), a conditional version of MRFs that was developed to address the limitations (e.g., local dependency and label bias) of directed models for sequential data (e.g., hidden Markov models and other discriminative Markov models based on directed graphical models). However, the inference and learning of general MRFs are challenging due to the presence of a global normalizing factor, i.e. partition function, especially when latent variables are present. Extensive efforts have been devoted to developing approximate methods. On one hand, sample-based methods (Neal, 1993) and variational approaches (Jordan et al., 1999; Welling & Sutton, 2005; Salakhutdinov & Larochelle, 2010) are proposed to infer the latent variables. On the other hand, extensive work (Meng & Wong, 1996; Neal, 2001; Hinton, 2002; Tieleman, 2008; Wainwright et al., 2005; Wainwright & Jordan, 2006) has been done to estimate the partition function. Among these methods, contrastive divergence (Hinton, 2002) is proven effective in certain types of models.

Most of the existing methods highly depend on the model structure and require model-specific analysis in new applications, which makes it important to develop black-box inference and learning methods. Previous work (Ranganath et al., 2014; Schulman et al., 2015) shows the ability to automatically infer the latent variables and obtain gradient estimate in directed models. However, there is no black-box learning method for undirected models except the recent work of NVIL (Kuleshov & Ermon, 2017).

NVIL introduces a variational distribution and derives an upper bound of the partition function in a general MRF, in the same spirit as amortized inference (Kingma & Welling, 2013; Rezende et al., 2014; Mnih & Gregor, 2014) for directed models. NVIL has several advantages over existing methods, including the ability of black-box learning, tracking the partition function during training and getting approximate samples efficiently during testing. However, NVIL also comes with two disadvantages: (1) it leaves the inference problem of MRFs unsolved[1] and only trains simple MRFs with tractable

---

[∗]Dept. of Comp. Sci. & Tech., BNRist Center, Institute for AI, THBI Lab, Tsinghua University, Beijing, 100084, China

[†]University of Amsterdam, and the Canadian Institute for Advanced Research (CIFAR).

[1]NVIL (Kuleshov & Ermon, 2017) presents a hybrid model. The "inference" in the title refers to directed part but not for an MRF.

posteriors, and (2) the upper bound of the partition function can be underestimated (Kuleshov & Ermon, 2017), resulting in sub-optimal solutions on high-dimensional data.

We propose *Adversarial Variational Inference and Learning* (AdVIL) to relieve some headache of learning an MRF model. AdVIL is a black-box inference and learning method that partly solves the two problems of NVIL and retains the advantages of NVIL at the same time. First, AdVIL introduces a variational encoder to infer the latent variables, which provides an upper bound of the free energy. Second, AdVIL introduces a variational decoder for the MRF, which provides a lower bound of the log partition function. The two variational distributions provide an estimate of the negative log-likelihood of the MRF. On one hand, the estimate is in an intuitive form of an approximate *contrastive free energy*, which is expressed in terms of the expected energy and the (conditional) entropy of the corresponding variational distribution. On the other hand, similar to GAN (Goodfellow et al., 2014), the estimate is a minimax optimization problem, which is solved by stochastic gradient descent (SGD) in an alternating manner. Theoretically, our algorithm is convergent if the variational decoder approximates the model well. This motivates us to introduce an auxiliary variable to enhance the flexibility of the variational decoder, whose entropy is approximated by the third variational trick.

We evaluate AdVIL in various undirected generative models, including restricted Boltzmann machines (RBM) (Ackley et al., 1985), deep Boltzmann machines (DBM) (Salakhutdinov & Hinton, 2009), and Gaussian restricted Boltzmann machines (GRBM) (Hinton & Salakhutdinov, 2006), on several real datasets. We empirically demonstrate that (1) compared to the black-box NVIL (Kuleshov & Ermon, 2017) method, AdVIL provides a tighter estimate of the log partition function and achieves much better log-likelihood results; and (2) compared to contrastive divergence based methods (Hinton, 2002; Welling & Sutton, 2005), AdVIL can deal with a broader family of MRFs without model-specific analysis and obtain better results when the model structure gets complex as in DBM.

## 2 BACKGROUND

We consider a general case where the model consists of both visible variables $v$ and latent variables $h$. An MRF defines the joint distribution over $v$ and $h$ as $P(v, h) = \frac{e^{-\mathcal{E}(v,h)}}{\mathcal{Z}}$, where $\mathcal{E}$ denotes the associated energy function that assigns a scalar value for a given configuration of $(v, h)$ and $\mathcal{Z}$ is the partition function such that $\mathcal{Z} = \int_{v,h} e^{-\mathcal{E}(v,h)} dv dh$.

Let $P_{\mathcal{D}}(v)$ denote the empirical distribution of the training data. Minimizing the negative log-likelihood (NLL) of an MRF is a commonly chosen learning criterion and it is given by:

$$\mathcal{L}(\theta) := -\mathbb{E}_{P_{\mathcal{D}}(v)} \left[ \log \int_h \frac{e^{-\mathcal{E}(v,h)}}{\mathcal{Z}} dh \right], \tag{1}$$

where $\theta$ denotes the trainable parameters in $\mathcal{E}$. Further, the gradient of $\theta$ is:

$$\nabla_\theta \mathcal{L}(\theta) = \mathbb{E}_{P_{\mathcal{D}}(v)} [\nabla_\theta \mathcal{F}(v)] - \mathbb{E}_{P(v)} [\nabla_\theta \mathcal{F}(v)], \tag{2}$$

where $\mathcal{F}(v) = -\log \int_h e^{-\mathcal{E}(v,h)} dh$ denotes the free energy and the gradient in Eqn. (2) is the difference of the free energy in two phases. In the first *positive phase*, the expectation of the free energy under the data distribution is decreased. In the second *negative phase*, the expectation of the free energy under the model distribution is increased.

Unfortunately, both the NLL in Eqn. (1) and its gradient in Eqn. (2) are intractable in general for two reasons. First, the integral of the latent variables in Eqn. (1) or equivalently the computation of the free energy in Eqn. (2) is intractable. Second, the computation of the partition function in Eqn. (1) or equivalently the negative phase in Eqn. (2) is intractable.

**Variational inference.** Extensive work introduces deterministic approximations for the intractability of inference, including the mean-field approximation (Welling & Hinton, 2002; Salakhutdinov & Hinton, 2009), the Kikuchi and Bethe approximations (Welling & Sutton, 2005) and the recognition model approach (Salakhutdinov & Larochelle, 2010). In this line of work, the intractability of the partition function is addressed using Monte Carlo based methods.

**Contrastive free energy.** Contrastive divergence (CD) (Hinton, 2002) addresses the intractability of the partition function by approximating the negative phase in Eqn. (2) as follows:

$$\nabla_\theta \mathcal{L}(\theta) = \mathbb{E}_{P_{\mathcal{D}}(v)} [\nabla_\theta \mathcal{F}(v)] - \mathbb{E}_{P_{CD}(v)} [\nabla_\theta \mathcal{F}(v)], \tag{3}$$

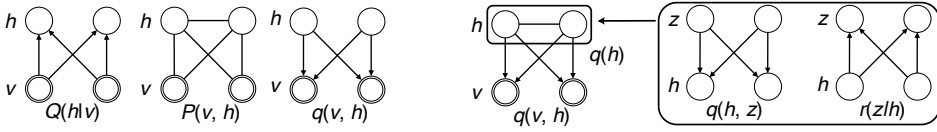

Figure 1: Illustration of the models involved in AdVIL. From left to right: variational encoder $Q(h|v)$, MRF $P(v, h)$, variational decoder $q(v, h)$ with a simple prior and $q(v, h)$ with an expressive prior.

where $P_{CD}(v)$ denotes the empirical distribution obtained by starting from a data point and running several steps of Gibbs sampling according to the model distribution and the free energy $\mathcal{F}(v)$ is assumed to be tractable. Existing methods (Welling & Hinton, 2002; Welling & Sutton, 2005) approximate $\mathcal{F}(v)$ using certain function $\mathcal{G}(v)$ and the gradient of $\theta$ is:

$$\nabla_\theta \mathcal{L}(\theta) \approx \mathbb{E}_{P_\mathcal{D}(v)} \left[ \nabla_\theta \mathcal{G}(v) \right] - \mathbb{E}_{P_{CD}(v)} \left[ \nabla_\theta \mathcal{G}(v) \right]. \tag{4}$$

Although these generalized methods exist, it is nontrivial to extend CD-based methods to general MRFs because the Gibbs sampling procedure is highly dependent on the model structure.

**Black-box learning.** The recent work of NVIL (Kuleshov & Ermon, 2017) addresses the intractability of the partition function in a black-box manner via a variational upper bound of the partition function:

$$\mathbb{E}_{q(v)} \left[ \frac{\tilde{P}(v)^2}{q(v)^2} \right] \geq \mathcal{Z}^2, \tag{5}$$

where $\tilde{P}(v) = e^{-\mathcal{F}(v)}$ is the unnormalized marginal distribution on $v$ and $q(v)$ is a neural variational distribution. As a black-box learning method, NVIL potentially allows application to broader model families and improves the capabilities of probabilistic programming systems (Carpenter et al., 2017). Though promising, NVIL leaves the intractability of inference in an MRF unsolved, and the bound in Eqn. (5) is of high variance and is easily underestimated (Kuleshov & Ermon, 2017).

## 3 METHOD

As stated above, the black-box inference and learning of MRFs are still largely open. In this paper, we make a step towards solving the problems by a new variational approach. For simplicity, we focus on the resulting objective function in this section. See Appendix A for detailed derivation.

### 3.1 ADVERSARIAL VARIATIONAL INFERENCE AND LEARNING

First, we rewrite the NLL of the MRF (See an illustration in Fig. 1) as follows:

$$\mathcal{L}(\theta) = -\mathbb{E}_{P_\mathcal{D}(v)} \left[ -\mathcal{F}(v) \right] + \log \mathcal{Z}, \tag{6}$$

where the negative free energy and the log partition function are in the form of a logarithm of an integral. Naturally, we can apply the variational trick (Jordan et al., 1999) twice and approximate the two terms individually. Due to the presence of the minus before the first term in Eqn. (6), the two variational tricks bound the two parts of the NLL in the opposite directions, detailed as below.

Formally, on one hand, we introduce an approximate posterior for the latent variables $Q(h|v)$, which is parameterized as a *neural variational encoder* (See an illustration in Fig. 1), to address the intractability of inference as follows:

$$\mathcal{L}(\theta) \leq \mathbb{E}_{P_\mathcal{D}(v)Q(h|v)} \left[ \mathcal{E}(v, h) + \log Q(h|v) \right] + \log \mathcal{Z} \coloneqq \mathcal{L}_1(\theta, \phi), \tag{7}$$

where $\phi$ denotes the trainable parameters in $Q(h|v)$. The upper bound is derived via applying the Jensen inequality and the equality holds if and only if $Q(h|v) = P(h|v)$ for all $v$. In the bound, the first term is the expected energy, which encourages $Q(h|v)$ to infer latent variables that have low values of the energy function $\mathcal{E}(v, h)$, or equivalently high probabilities of $P(v, h)$. The second term corresponds to the negative conditional entropy of $Q(h|v)$, which increases the uncertainty of $Q(h|v)$. In the paper, we denote the conditional entropy of $Q(h|v)$ as $\mathcal{H}(Q) \coloneqq -\mathbb{E}_{P_\mathcal{D}(v)Q(h|v)}[\log Q(h|v)]$.

On the other hand, we introduce an approximate sampler $q(v, h)$, which is parameterized by a *neural variational decoder* (See Fig. 1), to address the intractability of the partition function as follows:

$$\mathcal{L}_1(\theta, \phi) \geq \mathbb{E}_{P_{\mathcal{D}}(v)Q(h|v)}\left[\overbrace{\mathcal{E}(v, h)}^{\text{energy term}} + \overbrace{\log Q(h|v)}^{\text{entropy term}}\right]_{\text{Positive Phase}} - \mathbb{E}_{q(v,h)}\left[\overbrace{\mathcal{E}(v, h)}^{\text{energy term}} + \overbrace{\log q(v, h)}^{\text{entropy term}}\right]_{\text{Negative Phase}} := \mathcal{L}_2(\theta, \phi, \psi), \quad (8)$$

where $\psi$ denotes the trainable parameters in $q(v, h)$. The lower bound is derived via applying the Jensen inequality as well, and the equality holds if and only if $q(v, h) = P(v, h)$. It can be seen that the lower bound given by $q(v, h)$ consists of the entropy (denoted as $\mathcal{H}(q)$) and energy terms, which is similar to the upper bound in Eqn. (7), and the overall objective is in the form of approximate *contrastive free energy* (Hinton, 2002; Welling & Sutton, 2005). Because the double variational trick bounds the NLL in opposite directions as above, we have a minimax optimization problem:

$$\min_{\theta} \min_{\phi} \max_{\psi} \mathcal{L}_2(\theta, \phi, \psi). \quad (9)$$

The minimax formulation has been investigated in GAN (Goodfellow et al., 2014) and it is interpreted as an adversarial game between two networks. We name our framework *adversarial variational inference and learning* (AdVIL) following the well-established literature.

Note that $\mathcal{L}_2(\theta, \phi, \psi)$ is neither an upper bound, nor a lower bound of $\mathcal{L}(\theta)$ due to the double variational trick. However, we argue that solving the optimization problem in Eqn. (9) is reasonable because (1) it is equivalent to optimizing $\mathcal{L}(\theta)$ under the nonparametric assumption, which is similar to GAN (Goodfellow et al., 2014); and (2) it converges to a stationary point of $\mathcal{L}_1(\theta, \phi)$, which is an upper bound of $\mathcal{L}(\theta)$, under a weaker assumption, as stated in the following theoretical analysis.

## 3.2 THEORETICAL ANALYSIS OF AdVIL

In this section, we present our main theoretical results and the proofs can be found in Appendix C. Firstly, similarly to GAN (Goodfellow et al., 2014), we can prove that $\mathcal{L}_2$ is a tight estimate of $\mathcal{L}$ under the nonparametric assumption, which is summarized in Proposition 1 in Appendix C.1. However, the nonparametric assumption does not tolerate any approximation error between $P(v, h)$ and $q(v, h)$ during training and no guarantee can be obtained in finite steps. To this end, we establish a convergence theorem based on a weaker assumption that allows non-zero approximation error before convergence. A key insight is that the angle between $\frac{\partial \mathcal{L}_2(\theta, \phi, \psi)}{\partial \theta}$ and $\frac{\partial \mathcal{L}_1(\theta, \phi)}{\partial \theta}$ is positive if $q(v, h)$ approximates $P(v, h)$ well, as stated in the following Lemma 1.

**Lemma 1.** *For any $(\theta, \phi)$, there exists a symmetric positive definite matrix $H$ such that $\frac{\partial \mathcal{L}_2(\theta, \phi, \psi)}{\partial \theta} = H \frac{\partial \mathcal{L}_1(\theta, \phi)}{\partial \theta}$ under the assumption: $|| \sum_{v,h} \delta(v, h) \frac{\partial \mathcal{E}(v,h)}{\partial \theta} ||_2 < || \frac{\partial \mathcal{L}_1(\theta, \phi)}{\partial \theta} ||_2$ if $|| \frac{\partial \mathcal{L}_1(\theta, \phi)}{\partial \theta} ||_2 > 0$ and $|| \sum_{v,h} \delta(v, h) \frac{\partial \mathcal{E}(v,h)}{\partial \theta} ||_2 = 0$ if $|| \frac{\partial \mathcal{L}_1(\theta, \phi)}{\partial \theta} ||_2 = 0$, where $\delta(v, h) = q(v, h) - P(v, h)$.*

Based on Lemma 1 and other commonly used assumptions in the analysis of stochastic optimization (Bottou et al., 2018), AdVIL converges to a stationary point of $\mathcal{L}_1(\theta, \phi)$, as stated in Theorem 1.

**Theorem 1.** *Solving the optimization problem in Eqn. (9) using stochastic gradient descent, then $(\theta, \phi)$ converges to a stationary point of $\mathcal{L}_1(\theta, \phi)$ under the assumptions in the general stochastic optimization (Bottou et al., 2018) and that the condition of Lemma 1 holds in each step.*

Please see Appendix C.2 for a detailed and formal version of Theorem 1. Compared to Proposition 1 and the analysis in GAN (Goodfellow et al., 2014), Theorem 1 has a weaker statement that AdVIL converges to a stationary point of the negative evidence lower bound (i.e., $\mathcal{L}_1$) instead of $\mathcal{L}$. Nevertheless, we argue that converging to $\mathcal{L}_1$ is sufficiently good for variational approaches in general. Besides, Theorem 1 states that AdVIL can at least decrease $\mathcal{L}_1$ in expectation if the assumption holds in finite steps. Indeed, we empirically justify Theorem 1, as detailed in Appendix E.1. Theorem 1 also provides insights for the implementation of AdVIL. Indeed, its assumption motivates us to use a sufficiently powerful $q(v, h)$ with neural networks and auxiliary variables, and update $q(v, h)$ multiple times per update of $P(v, h)$, as detailed in Sec. 3.3 and Sec. 5.1 respectively.

## 3.3 SPECIFYING THE VARIATIONAL DISTRIBUTIONS

To efficiently get samples, both variational distributions are directed models. We use a directed neural network that maps $v$ to $h$ as the variational encoder $Q(h|v)$ (Kingma & Welling, 2013).

As for the variational decoder, we first factorize it as the product of a prior over $h$ and a conditional distribution, namely $q(v, h) = q(v|h)q(h)$. It is nontrivial to specify the prior $q(h)$ because the marginal distribution of $h$ in the MRF, i.e. $P(h)$, can be correlated across different dimensions. Consequently, a simple $q(h)$ is not flexible enough to track $P(h)$ and can violate the condition of Lemma 1. To this end, we introduce an auxiliary variable $z$, which can be discrete or continuous, on top of $h$ and define $q(v, h) = \int_z q(z)q(h|z)q(v|h)dz$.[2] (See an illustration in Fig. 1.) However, the entropy term of $q(v, h)$ is intractable because we need to integrate out the auxiliary variable $z$. Therefore, we introduce the third variational distribution $r(z|h)$ to approximate the entropy of $q(v, h)$. As in Eqn. (7), applying the standard variational trick gives an upper bound:

$$-\mathbb{E}_{q(v,h)} \log q(v, h) \leq -\mathbb{E}_{q(v,h)} \log q(v|h) - \mathbb{E}_{q(h)r(z|h)} \log \left[ \frac{q(h, z)}{r(z|h)} \right], \qquad (10)$$

which is unsatisfactory because the estimate is minimized w.r.t $r(z|h)$ while maximized w.r.t $q(v, h)$. Instead, after some transformations (See details in Appendix A) we get a lower bound as follows:

$$-\mathbb{E}_{q(v,h)} \log q(v, h) \geq -\mathbb{E}_{q(v,h)} \log q(v|h) - \mathbb{E}_{q(h,z)} \log \left[ \frac{q(h, z)}{r(z|h)} \right]. \qquad (11)$$

The equality holds if and only if $r(z|h) = q(z|h)$ for all $h$. The difference between the two bounds is subtle: the last expectation in Eqn. (10) is over $q(h)r(z|h)$ but that in Eqn. (11) is over $q(h, z)$. Here, a lower bound is preferable because the estimate is maximized with respect to both $r(z|h)$ and $q(v, h)$ and we can train them simultaneously. For simplicity, we absorb the trainable parameters of $r(z|h)$ into $\psi$. Note that after introducing $z$ and $r(z|h)$, we can still obtain a convergence theorem of AdVIL under the conditions that $r(z|h)$ approximates $q(z|h)$ well and $q(v, h) = \int q(v, h, z)dz$ is sufficiently close to $P(v, h)$ in every step, together the assumptions in general stochastic optimization.

Following GAN (Goodfellow et al., 2014), we optimize $\theta$, $\phi$ and $\psi$ jointly using stochastic gradient descent (SGD) in an alternating manner. The partial derivatives of $\phi$ and $\psi$ are estimated via the reparameterization trick (Kingma & Welling, 2013) for the continuous variables and the Gumbel-Softmax trick (Jang et al., 2016; Maddison et al., 2016) for the discrete variables. See Algorithm 1 in Appendix B for the whole training procedure. Note that $\psi$ is updated $K_1 > 1$ times per update of $\theta$.

## 4 RELATED WORK

Existing traditional methods (Neal, 2001; Hinton, 2002; Winn & Bishop, 2005; Wainwright & Jordan, 2006; Rother et al., 2007) can be used to estimate the log partition function but are nontrivial to be extended to learn general MRFs. Some methods (Winn & Bishop, 2005; Neal, 2001) require an expensive inference procedure for each update of the model and others (Hinton, 2002; Rother et al., 2007) cannot be directly applied to general cases (e.g., DBM). Among these methods, contrastive divergence (CD) (Hinton, 2002) is proven effective in certain types of models and it is closely related to AdVIL. Indeed, the partial derivative of $\theta$ in AdVIL is:

$$\frac{\partial \mathcal{L}_2(\theta, \phi, \psi)}{\partial \theta} = \mathbb{E}_{P_{\mathcal{D}}(v)Q(h|v)} \left[ \frac{\partial}{\partial \theta} \mathcal{E}(v, h) \right] - \mathbb{E}_{q(v,h)} \left[ \frac{\partial}{\partial \theta} \mathcal{E}(v, h) \right], \qquad (12)$$

which also involves a positive phase and a negative phase naturally and is quite similar to Eqn. (3). However, notably, the two phases average over the $(v, h)$ pairs and only require the knowledge of the energy function without any further assumption of the model in AdVIL. Therefore, AdVIL is more suitable to general MRFs than CD (See empirical evidence in Sec. 5.3).

In the context of black-box learning in MRFs, AdVIL competes directly with NVIL (Kuleshov & Ermon, 2017). It seems that the upper bound in Eqn. (5) is suitable for optimization because $P$ and $q$ share the same training direction. However, the bound holds only if the support of $\tilde{P}$ is a subset of the support of $q$. Further, the Monte Carlo estimate of the upper bound is of high variance. Therefore, the bound of NVIL can be easily underestimated, which results in sub-optimal solutions (Kuleshov & Ermon, 2017). In contrast, though AdVIL arrives at a minimax optimization problem, the estimate of Eqn. (8) is tighter and of lower variance. We empirically verify this argument (See Fig. 3) and systematically compare the two methods (See Tab. 1) in Sec.5.4.

---

[2]An alternative way is to use an autoregressive model as $q(h)$. See results and analysis in Appendix E.3.

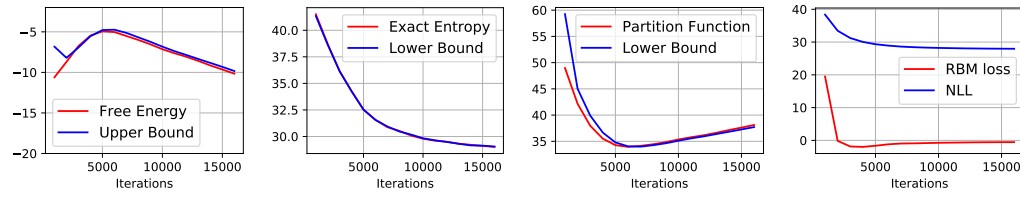

(a) upper bound of $\mathcal{F}(v)$    (b) lower bound of $\mathcal{H}(q)$    (c) lower bound of $\log \mathcal{Z}$    (d) RBM loss and NLL

Figure 2: Curves of AdVIL on Digits. (a-c) compare the values of the variational approximations and the corresponding ground truths. All bounds are rather tight after 5,000 iterations. (d) shows that the RBM loss (i.e., the loss of $\theta$ as in Eqn. (8)) tends to zero and the model converges gradually.

Apart from the work on approximate inference and learning in MRFs as mentioned above, AdVIL is also related to some directed models. Kim & Bengio (2016) jointly trains a deep energy model (Ngiam et al., 2011) and a directed generative model by minimizing the KL-divergence between them. Similar ideas have been highlighted in (Finn et al., 2016; Zhai et al., 2016; Dai et al., 2017; Liu & Wang, 2017). In comparison, firstly, AdVIL obtains the objective function in a unified perspective on the black-box inference and learning in general MRFs. Note that dealing with latent variables in MRFs is nontrivial (Kim & Bengio, 2016) and therefore existing work focuses on fully observable models. Secondly, AdVIL uses a sophisticated decoder with auxiliary variables to handle the latent variables and derives a principled variational approximation of the entropy term instead of the heuristics (Kim & Bengio, 2016; Zhai et al., 2016). Lastly, the convergence of AdVIL is formally characterized by Theorem 1 while the effect of the approximation error in inference is not well understood in existing methods. Adversarially learned inference (ALI) (Donahue et al., 2016; Dumoulin et al., 2016) is also formulated as a minimax optimization problem but focuses on directed models.

## 5   EXPERIMENTS

In this section, we evaluate AdVIL in restricted Boltzmann machines (RBM) (Ackley et al., 1985), deep Boltzmann machines (DBM) (Salakhutdinov & Hinton, 2009) and Gaussian restricted Boltzmann machines (GRBM) (Hinton & Salakhutdinov, 2006) on the Digits dataset, the UCI binary databases (Dheeru & Karra, 2017) and the Frey faces datasets (See detailed settings in Appendix D and the source code[3]). We compare AdVIL with strong baseline methods systematically and show the promise of AdVIL to learn a broad family of models effectively as a black-box method.

### 5.1   EMPIRICAL ANALYSIS OF ADVIL

We present a detailed analysis of AdVIL in RBM, whose energy function is defined as $\mathcal{E}(v, h) = -b^\top v - v^\top W h - c^\top h$. The conditional distributions of an RBM are tractable, but we still treat $P(h|v)$ as unknown and train AdVIL in a fully black-box manner. The analysis is performed on the Digits dataset and we augment the data of five times by shifting the digits following the protocol in (Kuleshov & Ermon, 2017). The dimensions of $v$, $h$ and $z$ are 64, 15 and 10, respectively. Therefore, the log partition function of the RBM and the entropy of the decoder can be computed by brute force.

Firstly, we empirically validate AdVIL in Fig. 2. Specifically, Panel (a) shows that the variational encoder $Q(h|v)$ provides a tight upper bound of the free energy after 2,000 iterations. Panel (b) demonstrates that the variational distribution $r(z|h)$ estimate the entropy of $q(v, h)$ accurately. Panel (c) shows that $q(v, h)$ can successfully track the log partition function after 5,000 iterations. Panel (d) presents that the RBM loss balances well between the negative phase and positive phase, and the model converges gradually. See Appendix E.1 for an empirical test of the condition in Lemma 1.

Secondly, we empirically show that both $P$ and $q$ can generate data samples in Appendix E.2.

Lastly, we analyze the sensitivity of $K_1$. Theoretically, enlarging $K_1$ will make $q(v, h)$ and $P(v, h)$ to be close and then help the convergence according to Theorem 1. As shown in Fig. 3 (a), a larger $K_1$ at least won't hurt the convergence, which agrees with Theorem 1. Though $K_1 = 15$ is sufficient on the Digits dataset, we use $K_1 = 100$ as a default setting for AdVIL on larger datasets.

---

[3]See the source code in https://anonymous.4open.science/r/8c779fbc-6394-40c7-8273-e52504814703/.

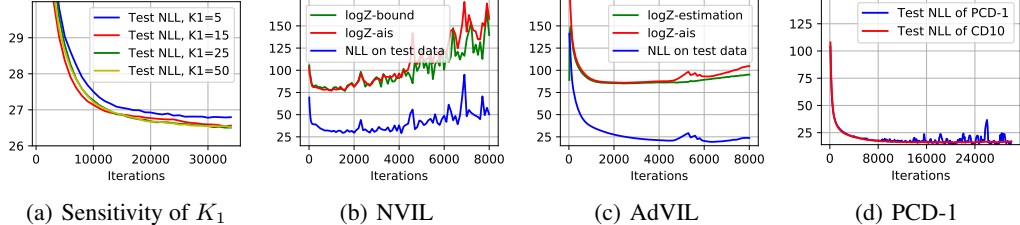

| (a) Sensitivity of $K_1$ | (b) NVIL | (c) AdVIL | (d) PCD-1 |

Figure 3: (a) Sensitivity analysis of $K_1$ on the Digits dataset. (b-d) Learning curves of NVIL, AdVIL and CD on the Mushrooms dataset. Compared to NVIL, AdVIL provides a tighter and lower variance estimate of $\log \mathcal{Z}$ and achieves better performance. Compared to PCD-1 and CD-10, AdVIL can track the log partition function and achieve comparable results though trained in a black-box manner.

Table 1: Anneal importance sampling (AIS) results in RBM. The results are recorded on the test set according to the best validation performance and averaged over three runs. AdVIL outperforms NVIL consistently and significantly. See the standard deviations in Appendix E.5.

| Method | Digits | Adult | Connect4 | DNA | Mushrooms | NIPS-0-12 | Ocr-letters | RCV1 |
|--------|--------|-------|----------|-----|-----------|-----------|-------------|------|
| NVIL-mean | $-27.36$ | $-20.05$ | $-24.71$ | $-97.71$ | $-29.28$ | $-290.01$ | $-47.56$ | $-50.47$ |
| AdVIL-mean | $\mathbf{-26.34}$ | $\mathbf{-19.29}$ | $\mathbf{-21.95}$ | $\mathbf{-97.59}$ | $\mathbf{-19.59}$ | $\mathbf{-276.42}$ | $\mathbf{-45.64}$ | $\mathbf{-50.22}$ |

## 5.2 RBM RESULTS

To the best of our knowledge, NVIL (Kuleshov & Ermon, 2017) is the only existing black-box learning method for MRFs and hence it is the most direct competitor of AdVIL. In this section, we provide a systematic comparison and analysis of these two methods in terms of the log-likelihood results on the UCI databases (Dheeru & Karra, 2017).

For a fair comparison, we use the widely-adopted anneal importance sampling (AIS) (Salakhutdinov & Murray, 2008) metric for quantitative evaluation. Besides, we carefully perform grid search over the default settings of NVIL (Kuleshov & Ermon, 2017) and our settings based on their code, and choose the best configuration including $K_1 = 100$ (See details in Appendix D). We directly compare with the best version of NVIL in Tab. 1. It can be seen that AdVIL consistently outperforms NVIL on all datasets, which demonstrate the effectiveness of AdVIL. Besides, the time complexity of AdVIL is comparable to that of NVIL with the same hyperparameters.

We compare the learning curves of NVIL and AdVIL on the Mushroom dataset. As shown in Fig. 3 (b), the upper bound of NVIL is underestimated after 4,000 iterations and then the model can get worse or even diverge. In contrast, as shown in Fig. 3 (c), the lower bound of AdVIL is consistently valid. Besides, the estimate of NVIL is looser and of higher variance than that of AdVIL. The results agree with our analysis in Sec. 4 and explain why AdVIL significantly outperforms NVIL. Further, as shown in Fig. 3 (d), AdVIL is comparable to CD-10 and persistent contrastive divergence (PCD) (Tieleman, 2008), which leverage the tractability of the conditional distributions in an RBM.

## 5.3 DBM RESULTS

We would like to demonstrate that AdVIL has the ability to deal with highly intractable models such as a DBM conveniently and effectively, compared to standard CD-based methods (Hinton, 2002; Welling & Hinton, 2002; Welling & Sutton, 2005) and NVIL (Kuleshov & Ermon, 2017).

DBM (Salakhutdinov & Hinton, 2009) is a powerful family of deep models that stack multiple RBMs together. The energy function of a two-layer DBM is defined as $\mathcal{E}(v, h_1, h_2) = -b^\top v - v^\top W_1 h_1 - c_1^\top h_1 - h_1^\top W_2 h_2 - c_2^\top h_2$. Learning a DBM is challenging because $P(h_1, h_2|v)$ is not tractable and CD (Hinton, 2002) is not applicable. Inspired by (Welling & Hinton, 2002; Welling & Sutton, 2005), we construct a variational CD (VCD) baseline by employing the same variational encoder $Q(h_1, h_2|v)$ as in AdVIL. The free energy is approximated by the same upper bound as in Eqn. (7), which is minimized with respect to the parameters in $Q(h_1, h_2|v)$. The gradient of the parameters in the DBM is given by Eqn. (4), where the Gibbs sampling procedure is approximated by $h_1 \sim Q(h_1|v)$ and $v \sim P(v|h_1)$. Note that AdVIL can be directly applied to this case. As for

Table 2: AIS results in DBM. The results are recorded according to the best validation performance and averaged by three runs. AdVIL achieves higher averaged AIS results on five out of eight datasets and has a better overall performance than VCD. See the standard deviations in Appendix E.5.

| Method | Digits | Adult | Connect4 | DNA | Mushrooms | NIPS-0-12 | Ocr-letters | RCV1 |
|---|---|---|---|---|---|---|---|---|
| VCD-mean | $-28.49$ | $-22.26$ | $-26.79$ | $\mathbf{-97.59}$ | $-23.15$ | $-356.26$ | $\mathbf{-45.77}$ | $\mathbf{-50.83}$ |
| AdVIL-mean | $\mathbf{-27.89}$ | $\mathbf{-20.29}$ | $\mathbf{-26.34}$ | $-99.40$ | $\mathbf{-21.21}$ | $\mathbf{-287.15}$ | $-48.38$ | $-51.02$ |

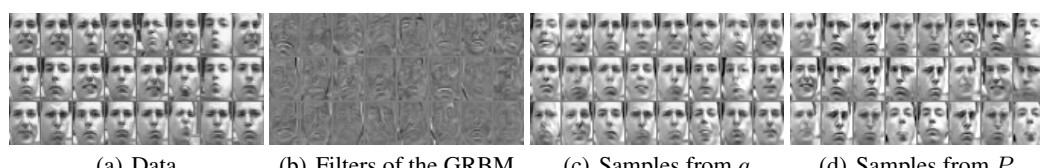

(a) Data       (b) Filters of the GRBM       (c) Samples from $q$       (d) Samples from $P$

Figure 4: Filters and samples of a GRBM learned by AdVIL on the Frey faces dataset. (a) presents the training data. (b) presents the first 40 filters of the GRBM. (c) and (d) show random samples from the variational decoder and the GRBM, respectively. We present the mean of $v$ for better visualization.

the time complexity, the training speed of AdVIL is around ten times slower than that of VCD in our implementation. However, the approximate inference and sampling procedure of AdVIL is very efficient thanks to the directed variational distributions.

The log-likelihood results on the UCI databases are shown in Tab. 2. It can be seen that AdVIL has a better overall performance even trained in a black-box manner, which shows the promise of AdVIL. See Appendix E.4 for learning curves and a detailed analysis of the results.

We also extend NVIL by using the same $Q(h_1, h_2|v)$ and $q(v, h_1, h_2)$ as AdVIL. However, NVIL diverges after 300 iterations and gets bad AIS results (e.g., less than $-40$ on Digits) in our implementation. A potential reason is that the upper bound given by $q$ in NVIL can be underestimated if $q$ is high-dimensional, as analyzed in Sec. 4 and Fig. 3. Note that $q(v, h_1, h_2)$ in DBM involves latent variables and has a higher dimension (e.g. 164 on the Digits dataset) than $q(v)$ in RBM (e.g. 64 on the Digits dataset). The results again demonstrate the advantages of AdVIL over NVIL.

## 5.4 GRBM RESULTS

We now show the ability of AdVIL to learn a GRBM on the continuous Frey faces dataset. The energy function of a GRBM is $\mathcal{E}(v, h) = \frac{1}{2\sigma^2}||v - b||^2 - c^\top h - \frac{1}{\sigma}v^\top W h$, where $\sigma$ is the standard deviation of the Gaussian likelihood and is set as 1 manually. We standardize the data by subtracting the mean and dividing by the standard deviation. The dimensions of $h$ and $z$ are 200 and 50, respectively.

Though a GRBM is more sensitive to the hyperparameters and hence harder to train than an RBM (Cho et al., 2011; 2013), AdVIL can successfully capture the underlying data distribution using the default hyperparameters (See Appendix D). As shown in Fig. 4, the samples from both the GRBM (via Gibbs sampling after 100,000 burn-in steps) and the decoder are meaningful faces. Besides, the filters of the GRBM outline diverse prototypes of faces, which accords with our expectation.

In summary, the results of the three models together demonstrate that AdVIL can learn a broad family of models conveniently and effectively in a fully black-box manner.

## 6 CONCLUSION AND DISCUSSION

A novel black-box learning and inference method for undirected graphical models, called adversarial variational inference and learning (AdVIL), is proposed. The key to AdVIL is a double variational trick that approximates the negative free energy and the log partition function separately. A formal convergence theorem, which provides insights for implementation, is established for AdVIL. Empirical results show that AdVIL can deal with a broad family of MRFs in a fully black-box manner and outperforms both the standard contrastive divergence method and the black-box NVIL algorithm.

Though AdVIL shows promising results, we emphasize that the black-box learning and inference of the MRFs are far from completely solved, especially on high-dimensional data. The two intractability

problems of MRFs are distinct since the posterior of the latent variables is *local* in terms of $v$ but the partition function is *global* by integrating out $v$. The additional integral makes estimating the partition function much more challenging. In AdVIL, simply increasing the number of updates of the decoder to obtain a tighter estimate of the partition function on high-dimensional data can be expensive. A potential future work to avoid the problem is adopting recent advances on non-convex optimization (Dauphin et al., 2014; Reddi et al., 2016; Wang et al., 2017) to accelerate the inner loop optimization. We conjecture that AdVIL is comparable to CD in RBM and superior to VCD in DBM on larger datasets if AdVIL can be trained to nearly converge based on our current results.

## ACKNOWLEDGEMENTS

This work was supported by the National Key Research and Development Program of China (No. 2017YFA0700904), NSFC Projects (Nos. 61620106010, U19B2034, U1811461), Beijing NSF Project (No. L172037), Beijing Academy of Artificial Intelligence (BAAI), Tsinghua-Huawei Joint Research Program, a grant from Tsinghua Institute for Guo Qiang, Tiangong Institute for Intelligent Computing, the JP Morgan Faculty Research Program and the NVIDIA NVAIL Program with GPU/DGX Acceleration. C. Li was supported by the Chinese postdoctoral innovative talent support program and Shuimu Tsinghua Scholar.

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

## A DERIVATION OF THE OBJECTIVE FUNCTION

Here we derive the objective function of AdVIL in detail. Let $\theta$, $\phi$ and $\psi$ denote the trainable parameters in the MRF, the variational encoder and the variational decoder, respectively. The first variational trick bounds the free energy as follows:

$$
\begin{aligned}
\mathcal{L}(\theta) &= -\mathbb{E}_{P_\mathcal{D}(v)} \left[ \log \left( \int_h e^{-\mathcal{E}(v,h)} dh \right) \right] + \log \mathcal{Z} \\
&= -\mathbb{E}_{P_\mathcal{D}(v)} \log \left[ \int_h Q(h|v) \frac{e^{-\mathcal{E}(v,h)}}{Q(h|v)} dh \right] + \log \mathcal{Z} \\
&\leq \mathbb{E}_{P_\mathcal{D}(v)Q(h|v)} \left[ \mathcal{E}(v,h) + \log Q(h|v) \right] + \log \mathcal{Z} \coloneqq \mathcal{L}_1(\theta, \phi),
\end{aligned}
$$

where the bound is derived via applying the Jensen inequality and the equality holds if and only if $Q(h|v) = P(h|v)$ for all $v$.

The second variational trick bounds the log partition function as follows:

$$
\begin{aligned}
\mathcal{L}_1(\theta, \phi) &= \mathbb{E}_{P_{\mathcal{D}}(v)Q(h|v)} \left[ \mathcal{E}(v, h) + \log Q(h|v) \right] + \log(\int_v \int_h e^{-\mathcal{E}(v,h)} dvdh) \\
&= \mathbb{E}_{P_{\mathcal{D}}(v)Q(h|v)} \left[ \mathcal{E}(v, h) + \log Q(h|v) \right] + \log(\int_v \int_h q(v,h) \frac{e^{-\mathcal{E}(v,h)}}{q(v, h)} dvdh) \\
&\geq \mathbb{E}_{P_{\mathcal{D}}(v)Q(h|v)} \left[ \mathcal{E}(v, h) + \log Q(h|v) \right] + \mathbb{E}_{q(v,h)} \left[ \log(\frac{e^{-\mathcal{E}(v,h)}}{q(v, h)}) \right] \\
&= \mathbb{E}_{P_{\mathcal{D}}(v)Q(h|v)} \left[ \underbrace{\overbrace{\mathcal{E}(v, h)}^{\text{energy term}} + \overbrace{\log Q(h|v)}^{\text{entropy term}}}_{\text{Positive Phase}} \right] - \mathbb{E}_{q(v,h)} \left[ \underbrace{\overbrace{\mathcal{E}(v, h)}^{\text{energy term}} + \overbrace{\log q(v,h)}^{\text{entropy term}}}_{\text{Negative Phase}} \right] \\
&:= \mathcal{L}_2(\theta, \phi, \psi),
\end{aligned}
$$

where the bound is also derived via applying the Jensen inequality and the equality holds if and only if $q(v, h) = P(v, h)$.

To enhance the expressive power of the variational decoder, we introduce an auxiliary variable $z$ and define $q(v, h) = \int_z q(z)q(h|z)q(v|h)dz$, which makes the entropy term in the negative phase intractable. To address the problem, we propose the third variational approximation. First, we can decompose the entropy of $q(v, h)$ as $-\mathbb{E}_{q(v,h)} \log q(v, h) = -\mathbb{E}_{q(v,h)} \log q(v|h) - \mathbb{E}_{q(h)} \log q(h)$ and we only need to approximate $-\mathbb{E}_{q(h)} \log q(h)$. However, simply applying the standard variational trick as above, we get an upper bound as follows:

$$
\begin{aligned}
-\mathbb{E}_{q(h)} \log q(h) &= -\mathbb{E}_{q(h)} \log \int_z q(h, z)dz \\
&= -\mathbb{E}_{q(h)} \log \int_z r(z|h) \frac{q(h, z)}{r(z|h)} dz \\
&\leq -\mathbb{E}_{q(h)r(z|h)} \log \left[ \frac{q(h, z)}{r(z|h)} \right],
\end{aligned}
$$

which is not satisfactory because the optimization problem will be $\min_P \min_Q \max_q \min_r$. Instead, we derive a lower bound as follows:

$$
\begin{aligned}
-\mathbb{E}_{q(h)} \log q(h) &= -\mathbb{E}_{q(h)} \log q(h) - \mathbb{E}_{q(h,z)} \log q(z|h) + \mathbb{E}_{q(h,z)} \log q(z|h) \\
&= -\mathbb{E}_{q(h,z)} \log q(h, z) + \mathbb{E}_{q(h,z)} \log q(z|h) \\
&= -\mathbb{E}_{q(h,z)} \log \left[ \frac{q(h, z)}{r(z|h)} \right] + \mathbb{D}_{KL}(q(z|h)||r(z|h)) \\
&\geq -\mathbb{E}_{q(h,z)} \log \left[ \frac{q(h, z)}{r(z|h)} \right],
\end{aligned}
$$

where $\mathbb{D}_{KL}(\cdot||\cdot)$ denotes the KL-divergence and the equality holds if and only if $r(z|h) = q(z|h)$ for all $h$. The difference between the two bounds is that the expectation is taken over $q(h)r(z|h)$ in the upper bound while over $q(h, z)$ in the lower bound. Using the lower bound, the optimization problem will be $\min_P \min_Q \max_p \max_r$.

## B    FORMAL TRAINING PROCEDURE

The formal training procedure of AdVIL is presented in Algorithm 1.

---

**Algorithm 1** Adversarial variational inference and learning by stochastic gradient descent

---

1: **Input:** Constants $K_1$ and $K_2$, learning rate schemes $\alpha$ and $\gamma$, randomly initialized $\theta$, $\phi$ and $\psi$
2: **repeat**
3:     **for** $i = 1, ..., K_1$ **do**
4:         Sample a batch of $(v, h, z) \sim q(v, h, z)$
5:         Estimate the objective of $q$ and $r$ according to Eqn. (11) and the negative phase in Eqn. (8)
6:         Update $\psi$ to maximize the objective according to $\alpha$
7:     **end for**
8:     **for** $i = 1, ..., K_2$ **do**
9:         Sample a batch of $(v, h) \sim P_{\mathcal{D}}(v)Q(h|v)$
10:        Estimate the objective of $Q$ according to the positive phase in Eqn. (8)
11:        Update $\phi$ to minimize the objective according to $\gamma$
12:     **end for**
13:     Sample a batch of $(v, h) \sim P_{\mathcal{D}}(v)Q(h|v)$ and another batch of $(v, h) \sim q(v, h)$
14:     Estimate the objective of $P$ according to Eqn. (8)
15:     Update $\theta$ to minimize the objective according to $\gamma$
16: **until** Convergence or reaching certain threshold

---

## C   DETAILED THEORETICAL ANALYSIS

For simplicity, we consider discrete $v$ and $h$ (e.g., in an RBM) and the analysis can be extended to the continuous cases. We assume $v \in \{0, 1\}^{d_v}$ and $h \in \{0, 1\}^{d_h}$, where $d_v$ and $d_h$ are the dimensions of the visible and latent variables respectively.

### C.1   ANALYSIS IN THE NONPARAMETRIC CASE

We first analyze the nonparametric case in Proposition 1 as follows.

**Proposition 1.** *For any $P(v, h) = \exp(-\mathcal{E}(v, h))/\mathcal{Z}$, $\mathcal{L}_2(\theta, \phi, \psi)$ is a tight estimate of the negative log-likelihood of $P(v)$, under the following assumptions*

    1. *$Q(h|v)$ and $q(v, h)$ are nonparametric.*

    2. *The inner optimization over $Q(h|v)$ and $q(v, h)$ can get their optima.*

*Proof.* Given $P(v, h)$, i.e., $\mathcal{E}(v, h)$, to find $q^*(v, h)$, we optimize $\mathcal{L}_2$ over $\{q(v, h)|v \in \{0, 1\}^{d_v}, h \in \{0, 1\}^{d_h}\}$ (we will use a shortcut $\{q(v, h)\}$ for simplicity). The optimization problem is equivalent to:

$$\min_{\{q(v,h)\}} \sum_{v,h} q(v, h) \left[\mathcal{E}(v, h) + \log q(v, h)\right]$$

$$\text{subject to: } \sum_{v,h} q(v, h) = 1,$$

$$q(v, h) \geq 0, \forall v, h.$$

Note that the objective function is convex since its Hessian matrix is positive semi-definite. Besides, the constraints are linear. Therefore, it is a convex optimization problem. Further, we can verify that the Slater's condition (Boyd & Vandenberghe, 2004) holds when $q$ is uniform and then the strong duality holds. Then, we can use the KKT conditions to solve the optimization problem.

The Lagrangian $\mathcal{G}(\{q(v, h)\}, \lambda, \{\mu(v, h)\})$ is:

$$\sum_{v,h} q(v, h) \left[\mathcal{E}(v, h) + \log q(v, h)\right] + \lambda(\sum_{v,h} q(v, h) - 1) + \sum_{v,h} \mu(v, h)q(v, h),$$

where $\lambda$ and $\{\mu(v, h)\}$ are the associated Lagrange multipliers.

To satisfy the stationarity, we take gradients with respect to $q(v, h)$ for all $(v, h)$ and get:

$$\left[\mathcal{E}(v, h) + \log q^*(v, h) + 1\right] + \lambda + \mu(v, h) = 0,$$

which implies

$$q^*(v, h) = \exp(-\mathcal{E}(v, h) - (1 + \lambda + \mu(v, h))).$$

According to the complementary slackness, we have

$$\mu(v, h)q^*(v, h) = 0, \forall v, h,$$

which implies $\mu(v, h) = 0, \forall v, h$, since $q^*(v, h) > 0, \forall v, h$.

To satisfy the primal equality constraint, we have

$$\sum_{v,h} q^*(v, h) = \sum_{v,h} \exp(-\mathcal{E}(v, h) - (1 + \lambda)) = 1,$$

which implies

$$q^*(v, h) = \frac{\exp(-\mathcal{E}(v, h))}{\sum_{v', h'} \exp(-\mathcal{E}(v', h'))} = P(v, h), \forall v, h.$$

To find $Q^*(h|v)$, we optimize $\mathcal{L}_2$ over $\{Q(h|v)|v \in \{0, 1\}^{d_v}, h \in \{0, 1\}^{d_h}\}$ (we will use a shortcut $\{Q(h|v)\}$ for simplicity). The optimization problem is equivalent to:

$$\min_{\{Q(h|v)\}} \sum_v P_{\mathcal{D}}(v) \sum_h Q(h|v) \left[\mathcal{E}(v, h) + \log Q(h|v)\right]$$

$$\text{subject to: } \sum_h Q(h|v) = 1, \forall v,$$

$$Q(h|v) \geq 0, \forall v, h.$$

Similar to the above procedure, we can get

$$Q^*(h|v) = \frac{\exp(-\mathcal{E}(v, h))}{\sum_{h'} \exp(-\mathcal{E}(v, h'))} = P(h|v), \forall v, h.$$

Under the assumptions that (1) $Q(h|v)$ and $q(v, h)$ are nonparametric, and (2) the inner optimization over $\psi$ and $\phi$ can get the optimum, the optimal variational distributions $P(v, h)$ and $P(h|v)$ can be obtained. Plugging them back into $\mathcal{L}_2$, we get

$$\mathcal{L}_2 = \mathbb{E}_{P_{\mathcal{D}}(v)P(h|v)} \left[\mathcal{E}(v, h) + \log P(h|v)\right] - \mathbb{E}_{P(v,h)} \left[\mathcal{E}(v, h) + \log P(v, h)\right]$$

$$= \mathbb{E}_{P_{\mathcal{D}}(v)P(h|v)} \left[-\log \sum_h e^{-\mathcal{E}(v,h)}\right] + \mathbb{E}_{P(v,h)} \left[\log \mathcal{Z}\right]$$

$$= \mathbb{E}_{P_{\mathcal{D}}(v)} \left[\mathcal{F}(v)\right] + \log \mathcal{Z} = \mathcal{L}.$$

$$\square$$

**Remark** Similar to Theorem 1 in (Goodfellow et al., 2014), Proposition 1 is under the nonparametric assumption, which is relaxed in our following analysis. Namely, we will consider more practical cases where $q(v, h)$ may not be exactly the same as $P(v, h)$ during training.

## C.2 MAIN CONVERGENCE THEOREM

For convenience, we summarize the training dynamics of Algorithm 1 with $K_1 = 1, K_2 = 1$ and the exact gradients (not the stochastic ones), as follows:

$$\psi_{k+1} = \psi_k + \alpha_k \frac{\partial \mathcal{L}_2(\theta_k, \phi_k, \psi_k)}{\partial \psi},$$

$$\phi_{k+1} = \phi_k - \gamma_k \frac{\partial \mathcal{L}_2(\theta_k, \phi_k, \psi_{k+1})}{\partial \phi},$$

$$\theta_{k+1} = \theta_k - \gamma_k \frac{\partial \mathcal{L}_2(\theta_k, \phi_k, \psi_{k+1})}{\partial \theta}, \tag{13}$$

where $k = 1, 2, \dots$. We will prove that even though we are optimizing $\mathcal{L}_2(\theta, \phi, \psi)$, $(\theta_k, \phi_k)$ converges to a stationary point of $\mathcal{L}_1(\theta, \phi)$ under certain conditions. To establish this, we first prove that the angle between $\frac{\partial \mathcal{L}_2(\theta, \phi, \psi)}{\partial \theta}$ and $\frac{\partial \mathcal{L}_1(\theta, \phi)}{\partial \theta}$ are sufficiently positive if $q(v, h)$ and $P(v, h)$ satisfy certain conditions, as summarized in Lemma 1.

**Lemma 1.** *For any $(\theta, \phi)$, there exists a symmetric positive definite matrix $H$ such that $\frac{\partial \mathcal{L}_2(\theta, \phi, \psi)}{\partial \theta} = H \frac{\partial \mathcal{L}_1(\theta, \phi)}{\partial \theta}$ under the assumption: $||\sum_{v,h} \delta(v,h) \frac{\partial \mathcal{E}(v,h)}{\partial \theta}||_2 < ||\frac{\partial \mathcal{L}_1(\theta, \phi)}{\partial \theta}||_2$ if $||\frac{\partial \mathcal{L}_1(\theta, \phi)}{\partial \theta}||_2 > 0$ and $||\sum_{v,h} \delta(v,h) \frac{\partial \mathcal{E}(v,h)}{\partial \theta}||_2 = 0$ if $||\frac{\partial \mathcal{L}_1(\theta, \phi)}{\partial \theta}||_2 = 0$, where $\delta(v,h) = q(v,h) - P(v,h)$.*

*Proof.* According to the Cauchy-Schwarz inequality, we have

$$\langle \frac{\partial \mathcal{L}_1(\theta, \phi)}{\partial \theta}, \sum_{v,h} \delta(v,h) \frac{\partial \mathcal{E}(v,h)}{\partial \theta} \rangle \leq ||\frac{\partial \mathcal{L}_1(\theta, \phi)}{\partial \theta}||_2 ||\sum_{v,h} \delta(v,h) \frac{\partial \mathcal{E}(v,h)}{\partial \theta}||_2.$$

If $||\frac{\partial \mathcal{L}_1(\theta, \phi)}{\partial \theta}||_2 > 0$, according to the assumption $||\sum_{v,h} \delta(v,h) \frac{\partial \mathcal{E}(v,h)}{\partial \theta}||_2 < ||\frac{\partial \mathcal{L}_1(\theta, \phi)}{\partial \theta}||_2$, we have

$$\langle \frac{\partial \mathcal{L}_1(\theta, \phi)}{\partial \theta}, \sum_{v,h} \delta(v,h) \frac{\partial \mathcal{E}(v,h)}{\partial \theta} \rangle < ||\frac{\partial \mathcal{L}_1(\theta, \phi)}{\partial \theta}||_2^2 = \langle \frac{\partial \mathcal{L}_1(\theta, \phi)}{\partial \theta}, \frac{\partial \mathcal{L}_1(\theta, \phi)}{\partial \theta} \rangle,$$

which implies that

$$\langle \frac{\partial \mathcal{L}_1(\theta, \phi)}{\partial \theta}, \frac{\partial \mathcal{L}_1(\theta, \phi)}{\partial \theta} - \sum_{v,h} \delta(v,h) \frac{\partial \mathcal{E}(v,h)}{\partial \theta} \rangle > 0.$$

According to the definitions of $\mathcal{L}_1(\theta, \phi)$ and $\mathcal{L}_2(\theta, \phi, \psi)$, we have

$$\frac{\partial \mathcal{L}_1(\theta, \phi)}{\partial \theta} - \sum_{v,h} \delta(v,h) \frac{\partial \mathcal{E}(v,h)}{\partial \theta}$$

$$= \sum_{v,h} [P_\mathcal{D}(v) Q(h|v) - P(v,h)] \frac{\partial \mathcal{E}(v,h)}{\partial \theta} - \sum_{v,h} [q(v,h) - P(v,h)] \frac{\partial \mathcal{E}(v,h)}{\partial \theta}$$

$$= \sum_{v,h} [P_\mathcal{D}(v) Q(h|v) - q(v,h)] \frac{\partial \mathcal{E}(v,h)}{\partial \theta} = \frac{\partial \mathcal{L}_2(\theta, \phi, \psi)}{\partial \theta},$$

which implies that

$$\langle \frac{\partial \mathcal{L}_1(\theta, \phi)}{\partial \theta}, \frac{\partial \mathcal{L}_2(\theta, \phi, \psi)}{\partial \theta} \rangle > 0.$$

Equivalently, there exists a symmetric positive definite matrix $H$ such that $\frac{\partial \mathcal{L}_2(\theta, \phi, \psi)}{\partial \theta} = H \frac{\partial \mathcal{L}_1(\theta, \phi)}{\partial \theta}$. Note that this also holds when $||\frac{\partial \mathcal{L}_1(\theta, \phi)}{\partial \theta}||_2 = 0$ (i.e., $\frac{\partial \mathcal{L}_1(\theta, \phi)}{\partial \theta} = \vec{0}$) because $||\frac{\partial \mathcal{L}_2(\theta, \phi, \psi)}{\partial \theta}||_2 \leq ||\frac{\partial \mathcal{L}_1(\theta, \phi)}{\partial \theta}||_2 + ||\sum_{v,h} \delta(v,h) \frac{\partial \mathcal{E}(v,h)}{\partial \theta}||_2 = 0$ (i.e., $\frac{\partial \mathcal{L}_2(\theta, \phi, \psi)}{\partial \theta} = \vec{0}$), according to the assumption. $\square$

**Remark** Lemma 1 assumes that $q(v,h)$ and $P(v,h)$ are sufficiently close, which is encouraged by choosing a sufficiently powerful family of $q(v,h)$ and updating $\psi$ multiple times per update of $\theta$, i.e. $K_1 > 1$. If Lemma 1 holds, optimizing $\mathcal{L}_2(\theta, \phi, \psi)$ with respect to $\theta$ will decrease $\mathcal{L}_1(\theta, \phi)$ in expectation with a sufficiently small stepsize. Also note that for any $(\theta, \psi)$, $\frac{\partial \mathcal{L}_2(\theta, \phi, \psi)}{\partial \phi} = \frac{\partial \mathcal{L}_1(\theta, \phi)}{\partial \phi}$ and therefore, optimizing $\mathcal{L}_2(\theta, \phi, \psi)$ with respect to $\phi$ will decrease $\mathcal{L}_1(\theta, \phi)$ in expectation with a sufficiently small stepsize.

Based on Lemma 1 and other commonly used assumptions in the analysis of stochastic gradient descent (Bottou et al., 2018), Algorithm 1 converges to a stationary point of $\mathcal{L}_1(\theta, \phi)$, as stated in Theorem 1.

**Theorem 1.** *Solving the optimization problem in Eqn. (9) using stochastic gradient descent according to Algorithm 1, then*

$$\lim_{k \to \infty} \mathbb{E}[||\frac{\partial \mathcal{L}_1(\theta_k, \phi_k)}{\partial \theta}||_2^2] = 0,$$

*under the following assumptions.*

1. *The condition of Corollary 4.12 in (Bottou et al., 2018): $\mathcal{L}_2(\theta, \phi, \psi)$ is twice differentiable with respect to $\theta$, $\phi$ and $\psi$.*

2. *Assumption 4.1 in (Bottou et al., 2018): the gradients of $\mathcal{L}_2(\theta, \phi, \psi)$ with respect to $\theta$, $\phi$ and $\psi$ are Lipschitz.*

3. *Assumption 4.3 in (Bottou et al., 2018): the first and second moments of the stochastic gradients are bounded by the expected gradients.*

4. *The stepsize satisfies the diminishing condition (Bottou et al., 2018), i.e., $\alpha_k = \gamma_k$, $\sum_{k=1}^{\infty} \gamma_k = \infty$, $\sum_{k=1}^{\infty} \gamma_k^2 < \infty$.*

5. *The condition of Lemma 1 holds in each step $k$. Therefore,*

$$\forall k, \exists H_k, \frac{\partial \mathcal{L}_2(\theta_k, \phi_k, \psi_{k+1})}{\partial \theta} = H_k \frac{\partial \mathcal{L}_1(\theta_k, \phi_k)}{\partial \theta}.$$

*Proof.* See Corollary 4.12 in (Bottou et al., 2018). □

**Remark** Assumption 1 and Assumption 2 in Theorem 1 are ensured because we use the sigmoid and tanh activation functions. Assumption 3 and Assumption 4 in Theorem 1 are ensured by the sampling and learning rate schemes of the stochastic gradient descent. Assumption 5 in Theorem 1 is weaker than the nonparametric assumption of Proposition 1 but still requires a large $K_1$. Also note that the statement of converging to $\mathcal{L}_1$ in Theorem 1 is weaker than that in Proposition 1.

### C.3 COMPLEMENTARY CONVERGENCE THEOREM

Heusel et al. (2017) propose a two-time scale update rule to train minimax optimization problems with a convergence guarantee even using $K_1 = 1$. AdVIL converges if using the same training method as in (Heusel et al., 2017), which is summarized in Proposition 2.

**Proposition 2.** *AdVIL trained with a two-time scale update rule (Heusel et al., 2017) converges to a stationary local Nash equilibrium almost surely under the following assumptions.*

1. *The gradients with respect to $\theta$, $\phi$ and $\psi$ are Lipschitz.*

2. $\sum_k \alpha_k = \infty$, $\sum_k \alpha_k^2 < \infty$, $\sum_k \gamma_k = \infty$, $\sum_k \gamma_k^2 < \infty$, $\gamma_k = o(\alpha_k)$.

3. *The stochastic gradient errors are bounded in expectation.*

4. *For each $\theta$, the ordinary differentiable equation corresponding to Equation 13 has a local asymptotically stable attractor within a domain of attraction such that the attractor is Lipschitz. Similar assumptions are required for $\phi$ and $\psi$.*

5. $\sup_k ||\theta_k|| < \infty$, $\sup_k ||\psi_k|| < \infty$, $\sup_k ||\phi_k|| < \infty$.

*Proof.* See Theorem 1 in (Heusel et al., 2017). □

**Remark** Compared to Theorem 1, Proposition 2 ensures the convergence of AdVIL without assuming $q(v, h)$ is sufficiently close to $P(v, h)$ in each step. However, a two time-sclae update rule (Heusel et al., 2017) is required to satisfy Assumption 2 and extra weight decay terms are needed to satisfy Assumption 4. Further, the convergence point is not necessarily a stationary point of $\mathcal{L}_1(\theta, \phi)$.

## D DATASETS AND EXPERIMENTAL SETTINGS

We evaluate our method on the Digits dataset[4], the UCI binary databases and the Frey faces datasets[5]. The information of the datasets is summarized in Tab. 3. We implement our model using the TensorFlow (Abadi et al., 2016) library. In all experiments, $q$ and $r$ are updated 100 times per update of $P$ and $Q$, i.e. $K_1 = 100$ and $K_2 = 1$. We use the ADAM (Kingma & Ba, 2014) optimizer with the learning rate $\alpha = 0.0003$, the moving average ratios $\beta_1 = 0.5$ and $\beta_2 = 0.999$, and the batch size of 500. We use a continuous $z$ and the sigmoid activation function . All these hyperparameters are

---

[4]https://scikit-learn.org/stable/modules/generated/ sklearn.datasets.load_digits.html#sklearn.datasets.load_digits
[5]http://www.cs.nyu.edu/~roweis/data.html

Table 3: Dimensions of the visible variables and sizes of the train, validation and test splits.

| Datasets | # visible | Train | Valid. | Test |
|---|---|---|---|---|
| Digits | 64 | 1438 | 359 | - |
| Adult | 123 | 5,000 | 1414 | 26147 |
| Connect4 | 126 | 16,000 | 4000 | 47557 |
| DNA | 180 | 1400 | 600 | 1186 |
| Mushrooms | 112 | 2,000 | 500 | 5624 |
| NIPS-0-12 | 500 | 400 | 100 | 1240 |
| OCR-letters | 128 | 32,152 | 10,000 | 10,000 |
| RCV1 | 150 | 40,000 | 10,000 | 150,000 |
| Frey faces | 560 | 1965 | - | - |

Table 4: The model structures in RBM experiments.

| | Digits | Adult | Connect4 | DNA | Mushrooms | NIPS-0-12 | Ocr-letters | RCV1 |
|---|---|---|---|---|---|---|---|---|
| dimension of $z$ | 15 | 15 | 15 | 15 | 15 | 50 | 15 | 15 |
| dimension of $h$ | 50 | 50 | 50 | 50 | 50 | 200 | 50 | 50 |
| dimension of $v$ | 64 | 123 | 126 | 180 | 112 | 500 | 128 | 150 |

set according to the validation performance of an RBM on the Digits dataset and fixed throughout the paper unless otherwise stated. The sizes of the variational distributions depend on the structure of the MRF and are chosen according to the validation performance. The model structures in RBM and DBM experiments are summarized in Tab. 4 and Tab. 5, respectively.

In a two-layer DBM with variables $v$, $h_1$ and $h_2$, we use an encoder $Q(h_1, h_2|v) = Q(h_1|v)Q(h_2|h_1)$ for both AdVIL and VCD. The decoder for AdVIL is the inverse of the encoder with one extra layer on the top, namely $q(v, h_1, h_2) = \int q(v|h_1)q(h_1|h_2)q(h_2|z)q(z)dz$. In our implementation, both AdVIL and VCD exploits that $v$ and $h_2$ are conditionally independent given $h_1$. The layer-wise structure potentially benefits the training of both methods. Nevertheless, in principle, any differentiable variational distributions be used in AdVIL and a systematical study is left for future work.

The authors of NVIL (Kuleshov & Ermon, 2017) propose two variants. The first one employs a mixture of Bernoulli as $q$. The second one involves auxiliary variables and employs a neural network as $q$. Both variants scale up to an RBM of at most 64 visible units as reported in their paper (Kuleshov & Ermon, 2017). For a fair comparison, we carefully perform grid search over the default settings of NVIL and our settings based on their code and choose the best configuration. In this setting, the first variant of NVIL still fails to scale up to larger datasets and the best version of the second variant shares the same key hyperparameters as AdVIL, including $K_1 = 100$ and a batch size of 500.

## E   MORE RESULTS

### E.1   EMPIRICAL VERIFICATION OF THEOREM 1

We empirically test the assumption $|| \sum_{v,h} \delta(v, h) \frac{\partial \mathcal{E}(v,h)}{\partial \theta_k} ||_2 < || \frac{\partial \mathcal{L}_1(\theta_k, \phi_k)}{\partial \theta_k} ||_2$ for $k < 10000$, where $\delta(v, h) = q(v, h) - P(v, h)$. Note that computing $|| \frac{\partial \mathcal{L}_1(\theta_k, \phi_k)}{\partial \theta_k} ||_2$ exactly requires summing over $v$ and $h$ and therefore we train a small RBM ,where the dimensions of $v$, $h$ and $z$ are 4 on a synthetic dataset. The data distribution is a categorical distribution over $\{0, 1\}^4$ and it is sampled from a Dirichlet distribution with all concentration parameters to be one. We get 10,000, 1,000 and 1,000 i.i.d samples from the categorical distribution for training, validation and test respectively. We find $K_1 = 10$ is sufficient in this case.

Table 5: The model structures in DBM experiments.

| | Digits | Adult | Connect4 | DNA | Mushrooms | NIPS-0-12 | Ocr-letters | RCV1 |
|---|---|---|---|---|---|---|---|---|
| dimension of $z$ | 15 | 15 | 15 | 15 | 15 | 50 | 15 | 15 |
| dimension of $h_1$ | 50 | 50 | 50 | 50 | 50 | 200 | 50 | 50 |
| dimension of $h_2$ | 50 | 50 | 50 | 50 | 50 | 200 | 50 | 50 |
| dimension of $v$ | 64 | 123 | 126 | 180 | 112 | 500 | 128 | 150 |

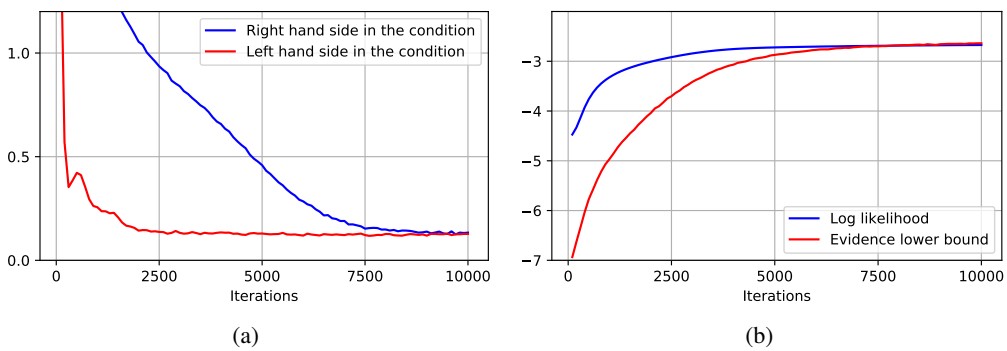

(a)  (b)

Figure 5: (a) shows that $||\sum_{v,h} \delta(v,h)\frac{\partial \mathcal{E}(v,h)}{\partial \theta_k}||_2$ (in red) is less than $||\frac{\partial \mathcal{L}_1(\theta_k, \phi_k)}{\partial \theta_k}||_2$ (in blue) during training. (b) shows that both the evidence lower bound (ELBO) $-\mathcal{L}_1$ (in red) and the log likelihood $-\mathcal{L}$ (in blue) converge gradually. The ELBO may be slightly over estimated because we approximate the first term in Eqn. (7) by a Monte Carlo estimate.

The results are shown in Fig. 5. It can be seen that a decoder with neural networks and auxiliary variables are sufficiently powerful to track the model distribution and therefore Assumption 5 in Theorem 1 holds in 10,000 steps. Besides, the model converges gradually, which agrees with our convergence analysis. The gap between the red curve in Fig. 5 (a) and the horizontal axis can be further reduced by using a more powerful $q(v, h)$ and advanced optimization techniques for $q(v, h)$.

Theoretically, how Lemma 1 holds as the number of variables increases is not clear. Intuitively, we agree that it may get harder to satisfy this condition in a high-dimensional space. However, it is still possible with advanced optimization methods because the MRF is randomly initialized and learned gradually, and the variational decoder is trained to track the MRF (based on an old version of the decoder) after every update of the MRF. Empirically, computing both sides of the condition requires the value of the partition function, which is as hard as training the MRF. Therefore, verifying this condition during training in a high-dimensional case is highly nontrivial.

### E.2  SAMPLES IN RBM

We present the samples from the RBM $P$ and the decoder $q$ in Fig. 6. In this case, we set the number of the hidden units to $50$ and other settings remain the same as in Sec. 5.1. The first column demonstrates that the decoder is a good approximate sampler for the RBM. Note that the samples from the decoder are obtained from efficient ancestral sampling but those from the RBM is obtained by Gibbs sampling after 100,000 burn-in steps. The second column shows that if $\mathcal{H}(Q)$ is removed, both models collapse to a certain mode of the data. The third column shows that if $\mathcal{H}(q)$ is removed, both models fail to generate meaningful digits. These results demonstrate the importance of the entropy terms and the necessity of approximating $\mathcal{H}(q)$ in a principled way.

### E.3  AdVIL WITH AN AUTOREGRESSIVE PRIOR

Here we present the results of AdVIL with a neural autoregressive distribution estimator (NADE) (Larochelle & Murray, 2011) as the prior on the Digits dataset. We use the same RBM as in Sec. 5.1.

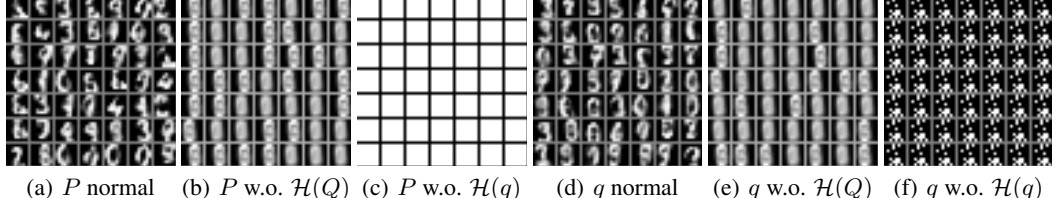

(a) $P$ normal    (b) $P$ w.o. $\mathcal{H}(Q)$ (c) $P$ w.o. $\mathcal{H}(q)$    (d) $q$ normal    (e) $q$ w.o. $\mathcal{H}(Q)$ (f) $q$ w.o. $\mathcal{H}(q)$

Figure 6: (a-c) Samples from the RBM in different settings. (d-f) Samples from the decoder in different settings. We present the mean of $v$ for better visualization in all settings.

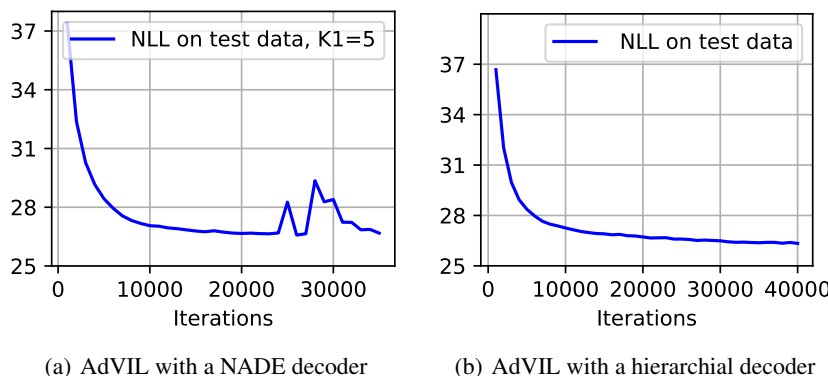

(a) AdVIL with a NADE decoder      (b) AdVIL with a hierarchial decoder

Figure 7: AdVIL with two types of decoders. We set $K_1 = 5$ in the NADE decoder and $K_1 = 100$ in the hierarchical decoder. The two models have a similar model capacity and training time.

The dimension of the latent units in NADE is 15, which is the same as the dimension of the auxiliary variables in the hierarchical decoder presented in Sec. 3.3.

Compared to the hierarchical decoder, the NADE decoder has a tractable entropy and hence does not require $r(z|h)$. However, getting samples from NADE is slow while AdVIL requires samples during training. Therefore, $K_1 = 5$ for the NADE decoder has a similar training cost as the hierarchical decoder.

Fig. 7 compares the two decoders. AdVIL with the NADE decoder achieves a slightly worse and unstable result.

### E.4 LEARNING CURVES AND ANALYSIS IN DBM

We plot the learning curves of AdVIL and VCD in DBM, as shown in Fig. 8. AdVIL achieves a better result than VCD, which agrees with the quantitative results in Tab. 2. Note that we report the test NLL results in Tab. 2 according to the best validation performance.

There are two types of biases introduced by using $Q(h_1, h_2|v) = Q(h_2|h_1)Q(h_1|v)$ in VCD. The first type of bias is introduced by using the approximate free energy in both the positive phase and the negative phase (See Eqn. (3) and Eqn. (4)). The second bias is introduced by the usage of $Q(h_1|v)$ to approximate $P(h_1|v)$ in the Gibbs sampling procedure to approximate the negative phase in Eqn. (4). The influence of the two types of biases on the negative phase is not clear, which can potentially explain the relatively inferior performance of VCD in DBM. In contrast, AdVIL approximates the negative phase by introducing one bias (i.e., the approximation error between $q(v, h)$ and $P(v, h)$), whose effect on learning is theoretically characterized by Theorem 1. The above results and analysis essentially demonstrate the advantages of AdVIL over CD-based methods in DBM and the importance of developing black-box inference and learning algorithms for general MRFs.

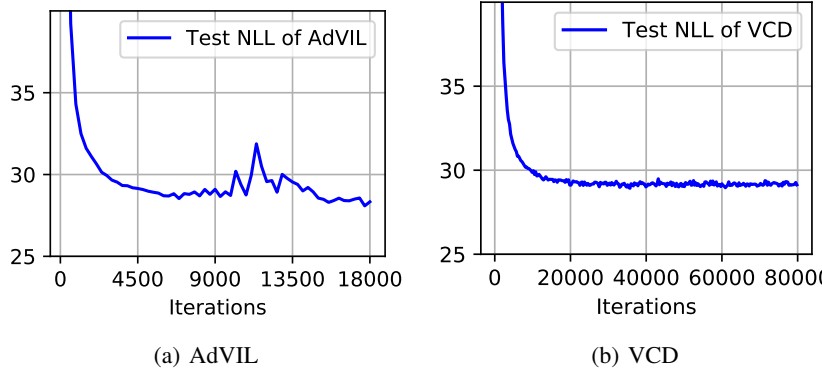

(a) AdVIL

(b) VCD

Figure 8: DBM results of AdVIL and VCD on the Digits dataset. The curve of AdVIL is less stable due to the presence of the minimax optimization problem but AdVIL achieves a better performance.

Table 6: The AIS results of NVIL and AdVIL in RBM with the means and standard deviations. The results are averaged over three runs with different random seeds.

| Method | Digits | Adult | Connect4 | DNA | Mushrooms | NIPS-0-12 | Ocr-letters | RCV1 |
|---|---|---|---|---|---|---|---|---|
| NVIL-mean | −27.36 | −20.05 | −24.71 | −97.71 | −29.28 | −290.01 | −47.56 | −50.47 |
| NVIL-std | 0.13 | 0.27 | 0.61 | 0.12 | 0.31 | 2.68 | 0.14 | 0.09 |
| AdVIL-mean | **−26.34** | **−19.29** | **−21.95** | **−97.59** | **−19.59** | **−276.42** | **−45.64** | **−50.22** |
| AdVIL-std | 0.02 | 0.07 | 1.04 | 0.10 | 2.01 | 0.21 | 0.34 | 0.06 |

## E.5    AIS RESULTS WITH STANDARD DEVIATIONS

The AIS results in RBM and RBM with the means and standard deviations are shown in Tab. 6 and Tab. 7 respectively.

Table 7: The AIS results of VCD-1 and AdVIL in DBM with the means and standard deviations. The results are averaged over three runs with different random seeds.

| Method | Digits | Adult | Connect4 | DNA | Mushrooms | NIPS-0-12 | Ocr-letters | RCV1 |
|---|---|---|---|---|---|---|---|---|
| VCD-mean | −28.49 | −22.26 | −26.79 | **−97.59** | −23.15 | −356.26 | **−45.77** | **−50.83** |
| VCD-std | 0.47 | 0.51 | 1.42 | 0.03 | 1.42 | 34.70 | 1.15 | 0.62 |
| AdVIL-mean | **−27.89** | **−20.29** | **−26.34** | −99.40 | **−21.21** | **−287.15** | −48.38 | −51.02 |
| AdVIL-std | 0.44 | 0.24 | 1.50 | 0.71 | 0.40 | 0.63 | 1.32 | 0.42 |

