# OpenReview forum: "To Relieve Your Headache of Training an MRF, Take AdVIL"
_ICLR.cc/2020/Conference — Accept (Poster)_

### Official Review · AnonReviewer3 · 2019-10-22
**Official Blind Review #3**

**Rating:** 6

**Review:**

This manuscript proposes a new approach to fitting Markov Random Fields (MRFs).  The general structure of the algorithm is amenable to many MRF structures and can be fairly straightforwardly applied to learning on a wide variety of problems.  The theoretical analysis supports that the algorithm is reasonable.  Experimental results show strong results on several different MRF models, albeit on relatively small problems.

I am giving this manuscript a weak accept.  The approach, to me, seems novel in fitting MRFs.  However, the theoretical claims and their limitations need to be more realistically discussed, and the empirical results need to be shown on larger and more complex datasets.

First, on the empirical results, there is a large literature on fitting RBM models, including many on scaling to much larger models.  Given that AdVIL actually diverges on the logZ estimation as the number of iterations goes up makes me worry about the efficacy of this approach on larger RBMs (Figure 2 uses v=64 and h=15 while a common RBM on MNIST is v=784 and h=500, a huge difference in scale).  As a large literature shows that estimating partition functions or normalizing constants gets much harder as the dimensionality goes up, I worry about this strategy and I think the manuscript would be greatly enhanced by looking at more common model sizes from the ML literature.

Also, to me, the classic MCMC+SGD is as much a black box as the proposed technique.  I realize that many of the top performing MCMC adapt specifically to the problem, but much of this is transferable between systems and can be put in simple sampling schemes.  Table 1 should be updated to include these typical techniques, because it is not clear that the proposed system actually outperforms the typical PCD-1 scheme (especially given in Figure 3).  Or succinctly, it should be made clear why I should use this over an MCMC approach.

To address these concerns, I would like the authors to answer how their proposed algorithm works in larger MRFs and do a more complete analysis compared to more traditional strategies.

Second, the theoretical claims are nice, but the manuscript should be revised to address the limitations of the theory.  In particular, Lemma 1 is extremely strong, and I disagree with the assessment that it is "much weaker" than the typical nonparametric assumption.  It seems that as the optimization gets close to the solution, this is essentially the exact same condition.  The authors need to clarify how exactly this is different, and dive into the practical implementations.  This also seems like it would get increasingly difficult as the number of hidden and visible units increases, so they should address how this Lemma holds as the theory scales.



**Experience Assessment:**

I have published in this field for several years.

**Review Assessment: Checking Correctness Of Derivations And Theory:**

I carefully checked the derivations and theory.

**Review Assessment: Checking Correctness Of Experiments:**

I carefully checked the experiments.

**Review Assessment: Thoroughness In Paper Reading:**

I read the paper thoroughly.

---

> ### Author Response · Authors · 2019-11-15
> **Response to Reviewer#3 (1/2)**
>
> We thank the reviewer for the positive and constructive comments. We revised the paper by providing extra discussion, experiments and analysis, and improving the presentation. We believe the quality of the paper is improved and we’d like to keep revising it if further feedbacks are provided.
>
> Q1: Experiments on larger datasets
>
> A1: Thanks for the suggestion. Indeed, we agree that estimating partition functions or normalizing constants gets much harder as the dimensionality goes up. For our method, as formally characterized by Theorem 1, the bottleneck to scaling up AdVIL is the inner loop to optimize $q(v, h)$. As discussed in Sec. 6, simply using large $K_1$ in our current strategy can be too expensive on high-dimensional data. A potential solution to avoid the problem is adopting recent advances on non-convex optimization (including but not limited to noise reduction methods [*1], saddle-free Newton methods [*2] and quasi-Newton methods [*3]) to accelerate the inner loop optimization, which is our important future work. We conjecture that AdVIL is comparable to CD and superior to VCD on larger datasets if AdVIL can be trained to nearly converge based on our results in Sec. 5.2 and Sec. 5.3.
>
> Besides, we argue that the main focus of this paper is to develop a flexible and well-understood inference and learning algorithm for general MRFs. The problem is significant as agreed by Reviewer#2 (with a forceful motivation presented in A2 to Reviewer#2). The problem is also largely open because a) CD and PCD require tractable conditional distributions; and b) VCD and methods based on layer-wise training [Salakhutdinov&Hinton 2009, Salakhutdinov&Larochelle 2010] are still model specific. Further, biases introduced in these methods are not well understood as well. AdVIL can be applied to different models easily and its convergence is formally characterized by Theorem 1. In the context of black-box inference and learning, we successfully scaled up to a scale of 200-500 in RBM and 200-200-500 in DBM. Note that the original paper of NVIL reported experiments up to a scale of 64-64 in RBM (much smaller than 200-500). The results in Sec. 5 showed the promise of AdVIL on dealing with different types of models compared to existing methods, as agreed by all reviewers. We hope the reviewers consider these conditions in the final decision. We added the above discussion in Sec. 6 according to the comment.
>
> Q2: Comparing with MCMC approach
>
> A2: We revised Sec. 4 to address the comments. MCMC and VI are complementary approaches for approximate inference. MCMC is asymptotically accurate but suffers from problems like slow mixing in practice. Estimating the log partition function accurately using MCMC is inefficient due to a large number of burn-in steps. For efficiency, advanced MCMC methods (rather than the block Gibbs sampling used in CD) can be combined with CD but this also introduces unclear bias. As representative ones of the MCMC methods, PCD and VCD are directly compared in this paper. As for the comparison to PCD, we argue that AdVIL does not necessarily outperform PCD in RBM because AdVIL makes a minimum assumption about the model structures while PCD requires tractable conditional distributions. Further, AdVIL outperforms VCD in DBM where the model structure is more complex than that in RBM.

---

> > ### Author Response · Authors · 2019-11-15
> > **Response to Reviewer#3 (2/2)**
> >
> >
> > Q3: The limitations of the theory
> >
> > A3: Thanks for the suggestion. We revised Sec. 3.2 and Appendix E to address the comment. We agree that Lemma 1 assumes that the variational decoder should recover the distribution of the MRF *at convergence*. However, the assumption is weaker in the sense that it allows non-zero approximation error between $P(v, h)$ and $q(v, h)$ *before convergence*. In contrast, the nonparametric assumption does not tolerate any approximation error in each step of the training. We argue that the difference is crucial in practice because Lemma 1 guarantees that AdVIL can at least increase the evidence lower bound ($-\mathcal{L}_1$) in expectation with a sufficiently small stepsize if the assumption holds in finite steps. Such a guarantee cannot be obtained by the analysis in GAN because using stochastic gradient descent to train $q(v, h)$ cannot ensure $P(v, h) = q(v, h)$ in finite steps. Indeed, we trained a small RBM to empirically verify Theorem 1 in Appendix E. 1. Fig. 5 shows that the condition of Lemma 1 holds in 10,000 steps and both the log likelihood ($-\mathcal{L}$) and the evidence lower bound ($-\mathcal{L}_1$) converge gradually. These results agree with our theoretical analysis.
> >
> > Theoretically, how this Lemma holds as the number of variables increases is not clear. Intuitively, we agree that it may get harder to satisfy this condition in a high-dimensional space. However, it is still possible with advanced optimization methods because the MRF is randomly initialized and learned gradually, and the variational decoder is trained to track the MRF (based on an old version of the decoder) after every update of the MRF. Empirically, computing both sides of the condition requires the value of the partition function, which is as hard as training the MRF. Therefore, verifying this condition during training in a high-dimensional case is highly nontrivial.
> >
> > [*1] Reddi S J, Hefny A, Sra S, et al. Stochastic variance reduction for nonconvex optimization[C]. International conference on machine learning. 2016: 314-323.
> > [*2] Dauphin Y N, Pascanu R, Gulcehre C, et al. Identifying and attacking the saddle point problem in high-dimensional non-convex optimization[C]. Advances in neural information processing systems. 2014: 2933-2941.
> > [*3] Wang X, Ma S, Goldfarb D, et al. Stochastic quasi-Newton methods for nonconvex stochastic optimization [J]. SIAM Journal on Optimization, 2017, 27(2): 927-956.

---

### Official Review · AnonReviewer1 · 2019-10-23
**Official Blind Review #1**

**Rating:** 6

**Review:**

The work proposes using variational distributions to model the model the inference of latent variables and model the partition function building on NVIL, thereby providing an algorithm that would work on general MRFs for both inference and learning. Since the two terms in the NLL are opposite in sign, it is a minimax operation and GAN like adversial training can be used. The paper shows providing tighter results to estimate the log partition function and comparisons on the digits dataset and Anneal importance sampling.

The paper builds on NVIL by using two variational distributions for the NLL and how to solve the parameter estimation problem. I think this strategy can be tested more extensively on more types of general MRFs and more rigourous experimentation and that the community will benefit from reading from these ideas. The paper contains some theory behind the work and experimental analysis of Advil including the sensitivity to parameters. Advil shows promise compared to the competing methods in some of the problems.

Questions and suggestions for improving the paper
1. Is this applicable to multivalued nodes or just binary problems? Or are any modifications needed?
2. Inference can be done in general using approximate methods like variational message passing, QBPO among others that don't depend on graph structures either, how does this work compared when those algorithms are used with simple gradient descent while training the parameters?
3. The paper states that the algorithm is convergent if the variational encoder approximates the model well. How would you define good approximation?
4. It would be good to add more details of how the GAN framework/adversial training is used.
5. Would it be possible to compare Advil to ALI in some of the experiments?
6. Fig. 2 d was not clear to me as why to expect the plot we see. The NLL flattens out with no progress.
7. It would be helpful to the reader to understand the comparison using persistent contrastive divergence. Why is it not used in other experiments. The paper says the main comparison point is NVIL but different experiments either mention ALI, VCD or PCD which is confusing.
8. It would be nice to see the time comparison between this learning parameters and other methods.





**Experience Assessment:**

I have read many papers in this area.

**Review Assessment: Checking Correctness Of Derivations And Theory:**

I assessed the sensibility of the derivations and theory.

**Review Assessment: Checking Correctness Of Experiments:**

I assessed the sensibility of the experiments.

**Review Assessment: Thoroughness In Paper Reading:**

I read the paper at least twice and used my best judgement in assessing the paper.

---

> ### Author Response · Authors · 2019-11-15
> **Response to Reviewer#1 (1/2)**
>
> We thank the reviewer for the positive and valuable comments. We revised the paper by providing more discussion and improving the presentation. We believe the quality of the paper is improved and we’d like to keep revising it if further feedbacks are provided.
>
> Q1: Is this applicable to multivalued nodes or just binary problems? Or are any modifications needed?
>
> A1: In principle, AdVIL can be applied to any kind of nodes, including the multivalued ones. If we have a different type of nodes, we need to modify the variational distributions of the corresponding nodes. For instance, categorical or multinomial distributions can be used for multivalued nodes. The general inference and learning algorithm remains the same.
>
> Q2: Inference can be done in general using approximate methods like variational message passing, QBPO among others that don't depend on graph structures either, how does this work compared when those algorithms are used with simple gradient descent while training the parameters?
>
> A2: We revised Sec. 4 to address the comment. Traditional methods including variational message passing and QBPO can estimate the log partition function of MRFs. However, using such methods with gradient descent to train parameters in general MRFs is highly nontrivial. On one hand, some methods e.g., variational message passing can be inefficient because they need to perform an expensive inference procedure for each update of the model. On the other hand, some methods e.g., QBPO may not be directly applied to latent variable models, especially DBM that learns hierarchical latent representations. These limitations of traditional methods motivate the development of black-box methods for general MRFs, built upon the recent advances on amortized inference.
>
> Q3: The paper states that the algorithm is convergent if the variational encoder approximates the model well. How would you define good approximation?
>
> A3: We corrected the typo in Sec. 1. AdVIL is convergent if the variational *decoder* (not the encoder) approximates the model well. The definition of “good approximation” is the condition for validity in Lemma 1, which assumes that the approximation error between $q(v, h)$ and $P(v, h)$ is bounded by the norm of the gradient of the negative evidence lower bound, i.e., $\mathcal{L}_1$. The approximation error is defined as the norm of the difference between the expected gradient over $P(v, h)$ and the expected gradient over $q(v, h)$.
>
> Q4: It would be good to add more details of how the GAN framework/adversarial training is used.
>
> A4: We clarified the issue in Sec. 3.1 to address the comment. As shown in Eqn. (9), AdVIL is formulated as a minimax optimization problem. The formulation has been investigated in GAN and interpreted as an *adversarial* game between two networks. We name our framework adversarial variational inference and learning following the well-established literature. The “adversarial training is used” in the sense that we also solve a minimax optimization problem.
>
> Q5: Would it be possible to compare AdVIL to ALI in some of the experiments?
>
> A5: Thanks for the suggestion. We revised Sec. 4 to clarify the difference. ALI focuses on the inference and learning for implicit *directed* models while this paper focuses on the inference and learning for *undirected* models. It is the future work to investigate whether the MRF in AdVIL can lead to a better decoder network compared to ALI.
>
> Q6: Fig. 2d was not clear to me as why to expect the plot we see. The NLL flattens out with no progress.
>
> A6: Fig. 2d shows that the NLL decreases quickly in 5,000 iterations and converges gradually (i.e., nearly but not exactly flattens out) in the following 10,000 iterations. Together with Fig. 2 (a-c), this verifies that all approximations in AdVIL work well in practice.

---

> > ### Author Response · Authors · 2019-11-15
> > **Response to Reviewer#1 (2/2)**
> >
> >
> > Q7: It would be helpful to the reader to understand the comparison using persistent contrastive divergence. Why is it not used in other experiments? The paper says the main comparison point is NVIL but different experiments either mention ALI, VCD or PCD which is confusing.
> >
> > A7: Thanks for the suggestion. ALI is not directly comparable to AdVIL (See the discussion in A5). In our experiments, we compared with NVIL (the most direct competitor) and CD-based methods (representative classical methods). NVIL can be applied to RBM and DBM and therefore it is compared in both settings. PCD and CD (newly added as required by Reviewer#2) are only compared in RBM because they are not applicable to DBM, which has an intractable $P(h|v)$. Instead, VCD (variational CD) is considered in DBM. Overall, as a black-box method, AdVIL outperforms NVIL in both settings and outperforms CD-based methods in DBM, showing the promise of AdVIL.
> >
> > Q8: It would be nice to see the time comparison between this learning parameters and other methods.
> >
> > A8: Thanks for the suggestion. We presented the time comparison with NVIL and VCD in Sec. 5.2 and 5.3, respectively. The time complexity of AdVIL is comparable to that of NVIL with the same hyperparameters. The training speed of AdVIL is around ten times slower than that of VCD. However, the approximate inference and sampling procedure of AdVIL is very efficient thanks to the directed variational distributions.

---

### Official Review · AnonReviewer2 · 2019-10-27
**Official Blind Review #2**

**Rating:** 6

**Review:**

This paper presents a black-box style learning algorithm for Markov Random Fields (MRF). The approach doubles down on the variational approach with variational approximations for both the positive phase and negative phase of the log likelihood objective function. For the negative phase, the authors use two separate variational approximations, one of which involves the modeling of the latent variable prior under the approximating distribution,

The approach is novel, as far as I know, though not particularly so, and I view this as one of the weak point of the paper. That said, it does seems like a fairly creative combination of existing approaches. As others have found in the past, a variational approximation to the partition function contribution to the loss function (i.e. the negative phase) results in the loss of the variational lower bound on log likelihood and the connection between the resulting approximation and the log likelihood becomes unclear. To deal with this issue, the authors argue (in Lemma 1) that the gradient of their approximate objective is at least in the same direction as the ELBO (lower bound) objective. The result is fairly obvious, but the conditions for validity have interesting consequences for the training algorithm, as it relates the approximation error to the norm of the gradient of the ELBO loss.

I have a minor issue with the discussion (in the last paragraph of sec. 3.2) stating that the theoretical statement of the proposed objective relies on a much weaker assumption than the nonparametric assumption made in the theoretical justification of GANs. While I agree with the statement as such, the GAN development makes a stronger statement about the nature of the learning trajectory. Specifically, it states that the generator is minimizing a Jenson-Shannon divergence which has a fixed point at the true data density. In the current development, Theorem 1 only states that the optimization process will converge to the stationary points of the approximate ELBO objective (L1 in the paper's notation).

Clarity: I found the paper to be very well written with a clear exposition of the material and sound development of the technical details.

Relevance and Significance: This paper is highly relevant to the ICLR community and -- to the extent that one believes that training and inference in MRFs is important -- also significant. One this last point, it seems ironic to me that the proposed strategy for training the MRF is through the use of three separate directed graphical models (an encoder q(h | x),  a decoder and a VAE to model the approximate prior over the latents h). In most modeling situations, one would simply impose the directed graphical model directly and skip the formalization in terms of an MRF. I would appreciate a more forceful motivation of the relevance of MRFs rather than just stating it as a important model with applications. What is unique
about the MRF formalism that -- for practical applications -- could not be effectively captured in a directed graphical model?
I note that I am aware of the theoretical representation differences between directed and undirected models, I am wondering how these differences actually matter in practical applications at scale.

Experiments: The authors show the empirical advantages offered by the proposed method over the existing literature. I was surprised not to see how this model performs on the binarized MNIST dataset, and would like to see that result as well as CIFAR likelihood.  MNIST, in particular, is a well studied dataset that many readers will be able to easily interpret. Its absence seems like a serious omission.

What is meant by "RBM loss" in Fig. 2(d), I do not see this defined?

I am somewhat alarmed at the use of 100 updates of the joint model q(v,h) (K1 = 100) for every update of the other parameters. For larger scale domains, I fear this could become an important obstacle to effective model training. The comparison to PCD-1 in Fig. 3 seems a bit unfair in that the learning curve ends at 8000 iterations, while PCD-1 continues to
improve NLL. I would like to see this curve extended until we start to see signs of overfitting. Perhaps PCD-1 results in performance that is far better than AdVIL. I would also like to see a comparison to CD-k, which often outperforms PCD-k. While I understand the stance taken by the authors that these methods leverage the tractability of the conditional
distributions, these strategies are sufficiently general to be considered widely applicable and a true competitor for AdVIL.

With respect to Deep Boltzmann Machine (DBM), I would prefer to see quantitative comparisons against published results. Here again, MNIST would be a useful dataset.  It seems as though, in the application of AdVIL to the DBM, the authors are exploiting the structure of the model in how they define their sampling procedure. Is that the case? More detail for this application of AdVIL would be nice. Also, I would like to see the test estimated NLL (via AIS) learning curves for VCD and AdVIL. Given the comparison to PCD in the RBM setting, I am somewhat surprised that AdVIL is so competitive with VCD in
the case of the DBM.


**Experience Assessment:**

I have published in this field for several years.

**Review Assessment: Checking Correctness Of Derivations And Theory:**

I carefully checked the derivations and theory.

**Review Assessment: Checking Correctness Of Experiments:**

I carefully checked the experiments.

**Review Assessment: Thoroughness In Paper Reading:**

I read the paper thoroughly.

---

> ### Author Response · Authors · 2019-11-15
> **Response to Reviewer#2 (1/2)**
>
> We thank the reviewer for the positive and constructive comments. We revised the paper by providing extra discussion and experiments and strengthening the motivation and clarification. We believe the quality of the paper is improved and we’d like to keep revising it if further feedback is provided.
>
> Q1: Discussion on the assumption and the theoretical statement in Sec. 3.2
>
> A1: We revised Sec. 3.2 and Appendix C to address the comment. Firstly, we clarified that Theorem 1 has a weaker statement (converging to $\mathcal{L}_1$ instead of $\mathcal{L}$) than that in GAN. We also emphasized that converging to $\mathcal{L}_1$ is sufficiently good for variational approaches in general. Please see additional discussion and empirical verification of Theorem 1 in A3 to Reviewer#3. Secondly, we analyzed AdVIL under the nonparametric assumption and obtained a similar statement to GAN. The analysis is formally summarized in Proposition 1 in Appendix C. 1.
>
> Q2: A forceful motivation of the relevance of MRFs in practical applications at scale
>
> A2: We strengthened the motivation in Sec. 1. In some practical applications such as image analysis [*1], social network analysis [*2] and biological informatics [*3], MRFs are preferable because the interactions between nodes (e.g., the relation between labels of adjacent pixels and friendship between people) are symmetric in nature. Directed models are not suitable for such data because of loops introduced by the directed edges in both directions.
>
> In particular, one famous example is Conditional Random Fields (Lafferty et al., 2001), a conditional version of MRFs that was developed to address the limitations (e.g., local dependency and label bias) of directed models for sequential data (e.g., Hidden Markov Models and other discriminative Markov models based on directed graphical models). CRFs have many successful applications in sequential data labeling and beyond.
>
> Q3: Experiments on larger datasets
>
> A3: Thanks for the suggestion. As formally characterized by Theorem 1, the bottleneck to scaling up AdVIL is the inner loop to optimize $q(v, h)$. As discussed in Sec. 6, simply using large $K_1$ in our current strategy is too expensive on high-dimensional data. A potential solution to avoid the problem is adopting recent advances on non-convex optimization (including but not limited to noise reduction methods [*4], saddle-free Newton methods [*5] and quasi-Newton methods [*6]) to accelerate the inner loop optimization, which is our important future work. We conjecture that AdVIL is comparable to CD in RBM and superior to VCD in DBM on larger datasets if AdVIL can be trained to nearly converge based on our results in Sec. 5.2 and Sec. 5.3.
>
> Besides, we argue that the main focus of this paper is to develop a flexible and well-understood inference and learning algorithm for general MRFs. The problem is significant as agreed by Reviewer#2 (with a forceful motivation presented in A2 to Reviewer#2). The problem is also largely open because a) CD and PCD require tractable conditional distributions; and b) VCD and methods based on layer-wise training [Salakhutdinov&Hinton 2009, Salakhutdinov&Larochelle 2010] are still model specific. Further, biases introduced in these methods are not well understood as well. AdVIL can be applied to different models easily and its convergence is formally characterized by Theorem 1. In the context of black-box inference and learning, we successfully scaled up to a scale of 200-500 in RBM and 200-200-500 in DBM. Note that the original paper of NVIL reported experiments up to a scale of 64-64 in RBM (much smaller than 200-500). The results on eight relatively small datasets in Sec. 5 showed the promise of AdVIL on dealing with different types of models compared to existing methods, as agreed by all reviewers. We hope the reviewers consider these conditions in the final decision. We added the above discussion in Sec. 6 according to the comment.
>
>
> Q4: “RBM loss” in Fig. 2 (d)
>
> A4: We clarified it in the caption of Fig. 2. The “RBM loss” refers to the loss of $\theta$ in Eqn. (8).
>
> Q5: The use of 100 updates of the joint model ($K_1 = 100$)
>
> A5: Please see some potential solutions to reduce $K_1$ or accelerate the inner optimization in general in A3.

---

> > ### Author Response · Authors · 2019-11-15
> > **Response to Reviewer#2 (2/2)**
> >
> >
> > Q6: The comparison to PCD-1 in Fig. 3
> >
> > A6: We updated the learning curve of PCD-1 (in blue) up to 30,000 iterations in Fig. 3 (d). The test NLL of PCD-1 is not stable since about 15,000 iterations, indicating that we have trained it for a sufficiently long time. The conclusion that AdVIL is comparable to PCD-1 still holds. We argue that AdVIL does not necessarily outperform PCD-1 and CD-k (added as required by Reviewer#2 in Q7) in RBM because  AdVIL makes a minimum assumption about the model structures while PCD-1 requires tractable conditional distributions. Further, AdVIL outperforms VCD in DBM where the model structure is more complex than that in RBM.
> >
> > Q7: The comparison to CD-k
> >
> > A7: We added the learning curve of CD-10 (in red) in Fig. 3 (d). CD-10 has a similar performance to PCD-1, which agrees with the results in [*7]. The same analysis as in A6 applies here.
> >
> > Q8: In DBM, the authors are exploiting the structure of the model in how they define their sampling procedure
> >
> > A8: We added some discussion in Appendix D to address the comment. In a two-layer DBM with variables $v$, $h_1$ and $h_2$, we use an encoder $Q(h_1, h_2|v) = Q(h_1|v) Q(h_2|h_1)$ for both AdVIL and VCD. The decoder for AdVIL is the inverse of the encoder with one extra layer on the top, namely $q(v, h_1, h_2) = \int q(v|h_1) q(h_1|h_2) q(h_2 |z) q(z) dz$. In our implementation, both AdVIL and VCD exploit that $v$ and $h_2$ are conditionally independent given $h_1$. The layer-wise structure potentially benefits the training of both methods. Nevertheless, in principle, any differentiable variational distributions be used in AdVIL and a systematic study is left for future work.
> >
> > Q9: Learning curves in DBM and why AdVIL is competitive to VCD in DBM
> >
> > A9: We added Appendix E. 4 to address the comment. We plotted the learning curves of AdVIL and VCD to train DBM in Fig. 8. AdVIL achieves a better result than VCD, which agrees with the quantitative results in Tab. 2. We emphasized that we trained both methods for a sufficiently long time (up to 80,000 iterations with an Adam optimizer) and reported the test NLL in Tab. 2 according to the best validation performance.
> >
> > There are two types of biases introduced by using $Q(h_1, h_2|v) = Q(h_2|h_1) Q(h_1|v)$ in VCD. The first type of bias is introduced by using the approximate free energy in both the positive phase and the negative phase (See Eqn. (3) and Eqn. (4)). The second bias is introduced by the usage of $Q(h_1|v)$ to approximate $P(h_1|v)$ in the Gibbs sampling procedure to approximate the negative phase in Eqn. (4). The influence of the two types of biases on *the negative phase* is not clear, which can potentially explain the relatively inferior performance of VCD in DBM. In contrast, AdVIL approximates the negative phase by introducing one bias (i.e., the approximation error between $q(v, h)$ and $P(v, h)$), whose effect on learning is theoretically characterized by Theorem 1. The above results and analysis essentially demonstrate the advantages of AdVIL over CD-based methods in DBM and the importance of developing black-box inference and learning algorithms for general MRFs.
> >
> >
> > [*1] Philipp Kr ̈ahenb ̈uhl and Vladlen Koltun.  Efficient inference in fully connected CRFs with Gaussian edge potentials. In Advances in neural information processing systems, pp. 109–117, 2011.
> > [*2] Newman M E J, Watts D J, Strogatz S H. Random graph models of social networks [J]. Proceedings of the National Academy of Sciences, 2002, 99(suppl 1): 2566-2572.
> > [*3] Wei Z, Li H. A Markov random field model for network-based analysis of genomic data [J]. Bioinformatics, 2007, 23(12): 1537-1544.
> > [*4] Reddi S J, Hefny A, Sra S, et al. Stochastic variance reduction for nonconvex optimization[C]. International conference on machine learning. 2016: 314-323.
> > [*5] Dauphin Y N, Pascanu R, Gulcehre C, et al. Identifying and attacking the saddle point problem in high-dimensional non-convex optimization[C]. Advances in neural information processing systems. 2014: 2933-2941.
> > [*6] Wang X, Ma S, Goldfarb D, et al. Stochastic quasi-Newton methods for nonconvex stochastic optimization [J]. SIAM Journal on Optimization, 2017, 27(2): 927-956.
> > [*7] Tieleman T. Training restricted Boltzmann machines using approximations to the likelihood gradient [C]. Proceedings of the 25th international conference on Machine learning. ACM, 2008: 1064-1071.

---

### Decision · Program_Chairs · 2019-12-19

**Decision:**

Accept (Poster)

**Comment:**

The paper proposes a black box algorithm for MRF training, utilizing a novel approach based on variational approximations of both the positive and negative phase terms of the log likelihood gradient (as R2 puts it, "a fairly creative combination of existing approaches").

Several technical and rhetorical points were raised by the reviewers, most of which seem to have been satisfactorily addressed, but all reviewers agreed that this was a good direction. The main weakness of the work is that the empirical work is very small scale, mainly due to the bottleneck imposed by an inner loop optimization of the variational distribution q(v, h). I believe it's important to note that most truly large scale results in the literature revolve around purely feedforward models that don't require expensive to compute approximations; that said, MNIST experiments would have been nice.

Nevertheless, this work seems like a promising step on a difficult problem, and it seems that the ideas herein are worth disseminating, hopefully stimulating future work on rendering this procedure less expensive and more scalable.